# Integration of human pancreatic islet genomic data refines regulatory mechanisms at Type 2 Diabetes susceptibility loci

Matthias Thurner[1,2], Martijn van de Bunt[1,2], Jason M Torres[1], Anubha Mahajan[1], Vibe Nylander[2], Amanda J Bennett[2], Kyle J Gaulton[3], Amy Barrett[2], Carla Burrows[2], Christopher G Bell[4,5], Robert Lowe[6], Stephan Beck[7], Vardhman K Rakyan[6], Anna L Gloyn[1,2,8], Mark I McCarthy[1,2,8]*

[1]The Wellcome Centre for Human Genetics, University of Oxford, Oxford, United Kingdom; [2]Oxford Centre for Diabetes, Endocrinology and Metabolism, University of Oxford, Oxford, United Kingdom; [3]Department of Pediatrics, University of California, San Diego, San Diego, United States; [4]Department of Twin Research and Genetic Epidemiology, Kings College London, London, United Kingdom; [5]MRC Lifecourse Epidemiology Unit, University of Southampton, Southampton, United Kingdom; [6]Centre for Genomics and Child Health, Blizard Institute, Barts and The London School of Medicine and Dentistry, London, United Kingdom; [7]Department of Cancer Biology, UCL Cancer Institute, University College London, London, United Kingdom; [8]Oxford NIHR Biomedical Research Centre, Churchill Hospital, Oxford, United Kingdom

*For correspondence:
mark.mccarthy@drl.ox.ac.uk

**Abstract** Human genetic studies have emphasised the dominant contribution of pancreatic islet dysfunction to development of Type 2 Diabetes (T2D). However, limited annotation of the islet epigenome has constrained efforts to define the molecular mechanisms mediating the, largely regulatory, signals revealed by Genome-Wide Association Studies (GWAS). We characterised patterns of chromatin accessibility (ATAC-seq, n = 17) and DNA methylation (whole-genome bisulphite sequencing, n = 10) in human islets, generating high-resolution chromatin state maps through integration with established ChIP-seq marks. We found enrichment of GWAS signals for T2D and fasting glucose was concentrated in subsets of islet enhancers characterised by open chromatin and hypomethylation, with the former annotation predominant. At several loci (including *CDC123, ADCY5, KLHDC5*) the combination of fine-mapping genetic data and chromatin state enrichment maps, supplemented by allelic imbalance in chromatin accessibility pinpointed likely causal variants. The combination of increasingly-precise genetic and islet epigenomic information accelerates definition of causal mechanisms implicated in T2D pathogenesis.
DOI: https://doi.org/10.7554/eLife.31977.001

## Introduction

T2D is a complex disease characterised by insulin resistance and reduced beta cell function. Recent GWAS have identified a large number of T2D susceptibility loci (*Scott et al., 2017*; *Mahajan et al., 2014*; *Wellcome Trust Case Control Consortium et al., 2012*; *Voight et al., 2010*), the majority of which affect insulin secretion and beta cell function (*Dimas et al., 2014*; *Wood et al., 2017*). However, most GWAS signals map to the non-coding genome and identification of the molecular

mechanisms through which non-coding variants exert their effect has proven challenging. Several studies have demonstrated that T2D-associated variants map disproportionately to regulatory elements, particularly those which influence RNA expression and cellular function of human pancreatic islets. (*Parker et al., 2013*; *Pasquali et al., 2014*; *van de Bunt et al., 2015*; *Olsson et al., 2014*; *Dayeh et al., 2014*; *Volkov et al., 2017*; *Varshney et al., 2017*; *Gaulton et al., 2015*, *2010*).

Characterisation of the islet regulome has until now been limited in scope. The use of DNA methylation and open chromatin data to further annotate ChIP-seq derived chromatin states has successfully uncovered novel biology for other diseases (*Wang et al., 2016*). Existing methylation studies in islets, however, have either profiled a very small proportion of methylation sites using methylation arrays (*Olsson et al., 2014*; *Dayeh et al., 2014*) or focused on T2D-associated disease differentially methylated regions (dDMRs) rather than the integration of DNA methylation status with T2D-relevant GWAS data (*Volkov et al., 2017*). At the same time, assays of open chromatin in human islets have been restricted to small sample numbers (limiting the potential to capture allelic imbalance in chromatin accessibility for example): these have focussed predominantly on the impact of clustered or 'stretch' enhancers (*Parker et al., 2013*; *Pasquali et al., 2014*; *Gaulton et al., 2010*; *Varshney et al., 2017*).

Most importantly, in part due to historical challenges in accessing human islet material or authentic human cellular models, reference annotations of the islet epigenome and transcriptome (in the context of projects such as GTEx, ENCODE and Epigenome Roadmap) have been largely absent. It is worth noting that islets constitute only ~1% of the pancreas, and islet epigenomes and transcriptomes cannot therefore be reliably assayed in analyses involving the entire organ. Previous islet epigenome studies have, therefore, had only limited ability to directly relate genetic variation to regulatory performance or to broadly characterise the role of DNA methylation in these processes.

In this study, we set out to expand upon previous studies of the islet regulome in several ways. First, we explored the human islet methylome in unprecedented depth using Whole-Genome Bisulphite Sequencing (WGBS) applied to a set of 10 human islet preparations. Second, we explored both basal and genotype-dependent variation in chromatin accessibility through ATAC-seq in 17 human islet samples. Third, we integrated these genome-wide data with existing islet regulatory annotations to generate a high-resolution, epigenome map of this key tissue. Finally, we used this detailed map to interpret GWAS signals for T2D (and the related trait of fasting glucose) and deduce the molecular mechanisms through which some of these loci operate.

## Results

### Characterising the DNA methylation landscape of human pancreatic islets

To characterise the human islet methylome and characterise the role of DNA methylation with respect to T2D genetic risk, we performed WGBS (mean coverage 13X) in human pancreatic islet DNA samples isolated from 10 non-diabetic cadaveric donors of European descent. Methylation levels across the genome were highly correlated across individual donors (mean CpG methylation Spearman's rho across 10 individual WGBS donors = 0.71, *Figure 1—figure supplement 1A*): we pooled the WGBS results to generate a single high-pass (mean coverage 85X) set of pooled human pancreatic islet methylation data covering 23.3 million CpG sites (minimum pooled coverage 10X).

Most previous studies of the relationship between GWAS data and tissue-specific methylation patterns (including those interrogating the relationship between islet methylation and T2D predisposition [*Dayeh et al., 2014*; *Olsson et al., 2014*]) had used data generated on the Illumina 450 k methylation array (*Hannon et al., 2016*; *Mitchell et al., 2016*; *Kato et al., 2015*; *Ventham et al., 2016*). For comparative purposes, we generated 450 k array methylation data from 32 islet samples ascertained from non-diabetic donors of European descent (five overlapping with those from whom WGBS data were generated). As with the WGBS data, methylation levels were highly correlated across individuals (mean CpG methylation Spearman's rho across 32 individual 450 k donor = 0.98, *Figure 1—figure supplement 1B*). After pooling 450 k array data across samples, methylation profiles generated from the 450 k array and WGBS were highly correlated at the small subset of total CpG sites for which they overlap: this was observed across pooled samples (pooled WGBS vs. 450 k

Spearman's rho = 0.89, *Figure 1A*) and across the five donors analysed by both methods (mean Spearman's rho = 0.80, not shown).

WGBS and 450 k array data differed substantially in terms of genome-wide coverage. The 450 k array was designed to interrogate with high precision and coverage ~480 k CpG sites (approximately 2% of all sites in the genome), selected primarily because they are located near gene promoters and CpG-island regions. The focus of the 450 k array on these regions, which tend to be less variable in terms of methylation, explains the high 450 k array correlation levels between donors. In addition, this selective design results in marked differences in the distributions of genome-wide methylation values between WGBS and the 450 k array. Whilst the WGBS data revealed the expected pattern of widespread high methylation levels with hypomethylation (<50%) restricted to 11.2% (2.6M/23.3M CpG sites) of the genome, the array disproportionately interrogated those hypomethylated sites (218 k [46%] of all 450 k CpG probes) (Kolmogorov–Smirnov (KS) test for difference, D = 0.40, $p<2.2\times10^{-16}$) (*Figure 1B*). These differences in methylation distribution were also evident within specific islet regulatory elements from previously defined standard chromatin state maps (*Parker et al., 2013*) (*Figure 1C*, *Figure 1—figure supplement 1C–D*). We found significant (FDR < 0.05) differences between the methylation levels of CpG sites accessed on the array, and those interrogated by WGBS, across most islet chromatin states: the largest differences were observed for weak promoters (median WGBS = 0.71 vs. median 450k = 0.11, KS test D = 0.51, $p<2.2\times10^{-16}$,) and weak enhancers (WGBS = 0.87 vs. 450 k median = 0.76, D = 0.39, $p<2.2\times10^{-16}$, *Figure 1—figure supplement 1D*).

In terms of coverage, most chromatin states, apart from promoters, were poorly represented by CpG sites directly interrogated by the array: for example the array assayed only ~2.9% of CpG sites in strong enhancer states (2.7–3.8% depending on strong enhancer subtype, *Figure 1C*). Although methylation levels were previously reported to be highly correlated across short (0.1–2 kb) genomic distances (*Zhang et al., 2015*; *Bell et al., 2011*; *Eckhardt et al., 2006*; *Guo et al., 2017*), the observed significant differences in the methylation distribution (*Figure 1C*, *Figure 1—figure supplement 1D*) across chromatin states including weak promoter (median size 600 bp) and enhancer subtypes (median size ranges from 200 to 1200 bp) indicate that these correlative effects are not strong enough to counterbalance the low coverage of the 450 k array. These findings are consistent with 450 k array content being focused towards CpG-dense hypomethylated and permissive promoter regions. This highlights the limited capacity of the array to comprehensively interrogate the global DNA methylome, in particular at distal regulatory states such as enhancers.

To understand the value of these data to reveal molecular mechanisms at GWAS loci, where we and others had shown enrichment for islet enhancer states (*Pasquali et al., 2014*; *Gaulton et al., 2015*; *Parker et al., 2013*), we were interested to see how the selective coverage of the array might impact on its ability to interrogate methylation in GWAS-identified regions. We used the largest currently available T2D DIAGRAM GWAS data set (involving 26.7 k cases and 132.5 k controls of predominantly European origin, see dataset section for details) to identify the 'credible sets' of variants at each locus which collectively account for 99% of the posterior probability of association (PPA) (*Scott et al., 2017*; *Maller et al., 2012*).

To estimate the respective proportions of these T2D-associated variants captured by CpG sites assayed by the 450 k array and WGBS, we determined, for each locus, the combined PPA of all 99% credible set variants mapping within 1000 bp of any CpG site captured. This is based on evidence that across short distances CpG methylation is highly correlated (*Zhang et al., 2015*; *Bell et al., 2011*; *Eckhardt et al., 2006*; *Guo et al., 2017*) and may be influenced by genetic variants associated with altered transcription factor binding (*Do et al., 2016*). We found that coverage of this space of putative T2D GWAS variants by the 450 k array is low: across GWAS loci, the combined PPA attributable to variants within regions assayed by the array ranged from 0–99% with a median PPA per locus of 16% (compared to a WGBS median PPA per locus = 99%, KS-test $p<2.2\times10^{-16}$, *Figure 1D*, top). We estimated that the equivalent figure for a recently developed upgrade of the 450 k array, which captures ~850 k CpG sites and aims to provide better coverage of enhancer regions, would be ~39% (range 0%–99% *Figure 1D*, top). For instance, at the *DGKB* T2D locus (centred on rs10276674), CpG sites covered by the 450 k array interrogated less than 1% of the PPA of associated variants (vs. 99% captured by WGBS); the figure for the 850 k array would be 23% (*Figure 1E*). We obtained similar results when we performed equivalent analyses using GWAS data

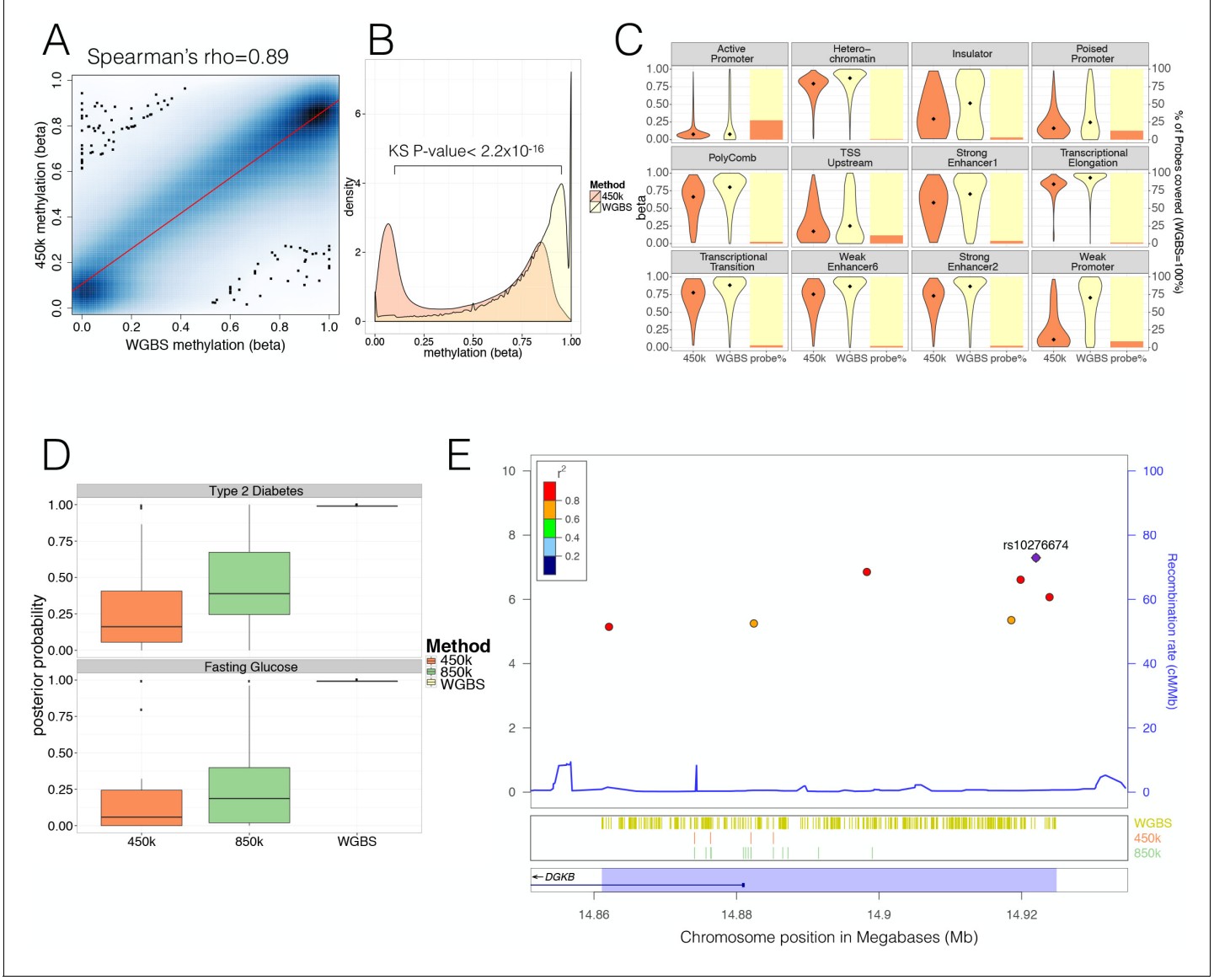

**Figure 1.** Comparison of human pancreatic islet WGBS and 450 k methylation data across the genome. (**A**) Smooth Scatter plot shows Spearman's rho correlation between the 450 k array (y-axis) and WGBS (x-axis) at overlapping sites. Darker colour indicates higher density of sites. (**B**) Comparison of the 450 k array (orange) and WGBS (yellow) methylation levels (x-axis) of all CpGs genome-wide assayed by either method (y-axis shows density). The P-value shown is derived using a Kolmogorov-Smirnov (KS) test. (**C**) For each chromatin state from *Parker et al. (2013)* the methylation levels of all CpG sites independent of overlap (diamond indicates the median) are shown as violin plots (left y-axis) and the CpG probe percentage per state for the 450 k array (orange) and WGBS (yellow) are shown as bar-plot (right y-axis). The 450 k probes represent the percentage of the total number of CpG sites which is determined by the number of WGBS CpG sites detected (WGBS = 100%). (**D**) Distribution of GWAS Posterior Probabilities (Type 2 Diabetes and Fasting Glucose) captured by CpG sites on the 450 k array (orange), 850 k array (green) and WGBS (yellow/black line). (**E**) Locuszoom plot showing CpG density and credible set SNPs. SNPs are shown with P-values (dots, y-axis left), recombination rate (line, y-axis right) and chromosome positions (x-axis) while CpG and gene annotations are shown below. These annotations include CpGs identified from WGBS (yellow stripes), 450 k CpG probes (orange stripes), 850 k CpG probes (green stripes) and gene overlap (*DGKB* label). The highlighted region in blue captures the 99% credible set region plus additional 1000 bp on either side. At the very bottom the position on chromosome seven is shown in Megabases (Mb).

DOI: https://doi.org/10.7554/eLife.31977.002

The following figure supplements are available for figure 1:

**Figure supplement 1.** Correlation of DNA methylation across WGBS and 450 k sites and comparison of WGBS and 450 k methylation levels across chromatin states.

DOI: https://doi.org/10.7554/eLife.31977.003

**Figure supplement 2.** PC analysis of 450 k DNA methylation samples.

DOI: https://doi.org/10.7554/eLife.31977.004

for fasting glucose (FG, from the ENGAGE consortium [*Horikoshi et al., 2015*]), another phenotype dominated by islet dysfunction (*Figure 1D*, bottom).

These data indicate that available methylation arrays provide poor genome-wide coverage of methylation status and are notably deficient in capturing methylation status around the distal regulatory enhancer regions most relevant to T2D predisposition. For this reason, we focused subsequent analyses on the WGBS data.

## Integration islet methylation and other epigenomic annotations

Studies in a variety of other tissues have shown that hypomethylation is a strong indicator of regulatory function (*Stadler et al., 2011*). More specifically, continuous stretches of CpG-poor Low-Methylated Regions (LMRs, with methylation ranging from 10–50% and containing fewer than 30 CpG sites) denote potential distal regulatory elements such as enhancers, while stretches of CpG-rich UnMethylated Regions (UMRs, containing more than 30 CpG sites) are more likely to represent proximal regulatory elements including promoters (*Burger et al., 2013*). We detected 37.1 k LMRs, 13.6 k UMRs (*Figure 2A*) and 10.7 k Partially Methylated Domains (PMDs) (Materials and methods and *Figure 2—figure supplement 1A–B*). PMDs represent large regions of unordered methylation states associated with DNA sequence features (*Gaidatzis et al., 2014*). As anticipated, we found significant

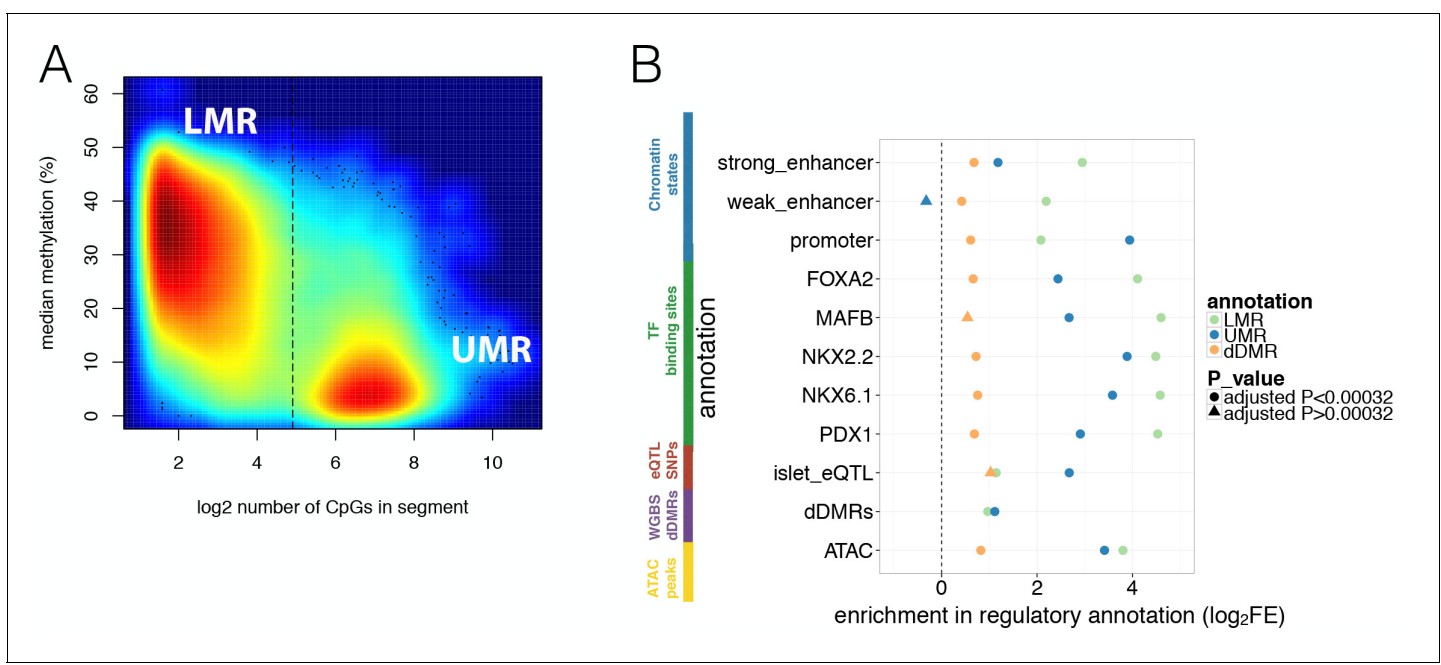

**Figure 2.** Overlap of WGBS hypomethylation and ATAC-seq open chromatin peaks with regulatory annotation. (**A**) Methylation levels in percent (y-axis) and $\log_2$ CpG density (x-axis) of UMR and LMR regulatory regions with the dashed line indicating the CpG-number (30 CpGs) that distinguishes LMRs and UMRs. (**B**) Log2 Fold Enrichment ($\log_2$FE) of LMRs (green shape), UMRs (blue shape) in various islet annotations is shown. These annotations include islet chromatin states, islet relevant TFBS (FOXA2, MAFB, NKX2.2, NKX6.1, PDX1), islet eQTLs, WGBS derived T2D-associated islet disease DMRs (dDMRs) and ATAC-seq open chromatin peaks. The dDMRs were derived from 6 T2D and eight non-diabetic individuals by *Volkov et al. (2017)* and dDMRs (orange shape) were also tested for enrichment in the aforementioned islet regulatory annotations. For all annotations, the empirically determined Bonferroni adjusted P-value is ≤0.00032 unless otherwise indicated by the shape: a dot corresponds to an Bonferroni adjusted p-value<0.00032 while the three triangles indicates Bonferroni adjusted p-values>0.00032: UMR enrichment adjusted P-value for weak enhancers = 1; dDMR enrichment adjusted P-value for MAFB = 0.006 and dDMR enrichment adjusted P-value for islet eQTLs = 0.01.
DOI: https://doi.org/10.7554/eLife.31977.005

The following source data and figure supplement are available for figure 2:

**Source data 1.** LMR_UMR_source_MThurner_Oct_2017.tds is associated with primary *Figure 2* Bed file providing coordinates of WGBS hypomethylated regulatory regions defined as UMRs and LMRs.
DOI: https://doi.org/10.7554/eLife.31977.007

**Figure supplement 1.** Identification and removal of Partially Methylated Domains (PMDs) and additional characterisation of regulatory regions.
DOI: https://doi.org/10.7554/eLife.31977.006

enrichment of LMRs with weak and strong enhancer states as defined by islet chromatin state maps derived from existing ChIP-seq data (*Parker et al., 2013*) (69.2% of islet LMRs overlapped islet strong and weak enhancer states, $\log_2$FE = 2.2–2.9, Bonferroni p<0.05, *Figure 2B*, *Figure 1—figure supplement 1C*). Similarly, UMRs were enriched for islet active promoter chromatin states (90.8% of UMRs overlapped islet active promoters, $\log_2$FE = 3.9, Bonferroni p < 0.05, *Figure 2B*).

To further characterise these hypomethylation domains, we overlapped information from analyses of islet cis-expression QTLs (eQTLs) (*van de Bunt et al., 2015*) and islet ChIP-seq transcription factor binding sites (TFBS) (*Pasquali et al., 2014*). We observed marked enrichment for eQTLs (LMR $\log_2$FE = 1.1, UMR $\log_2$FE = 2.7, Bonferroni p<0.05) and TFBS (LMR $\log_2$FE = 4.1–4.6; UMR $\log_2$FE = 2.4–3.9, Bonferroni p<0.05, *Figure 2B*). These observations confirm that islet LMRs and UMRs correspond to important tissue-specific regulatory regions, overlapping cis-regulatory annotations known to be enriched for T2D GWAS signals (*Pasquali et al., 2014*; *Gaulton et al., 2015*).

We also considered the relationship between LMR and UMR regions defined in our non-diabetic islet WGBS, and a complementary set of methylation-based annotations previously derived from WGBS of islets from 6 T2D and 8 control individuals (*Volkov et al., 2017*). In that study, comparisons between diabetic and non-diabetic islets had been used to define a set of 25,820 'disease differentially methylated regions' (dDMRs, minimum absolute methylation difference 5% and p<0.02). We found only limited overlap between these dDMRs and the UMRs and LMRs from our data: of the 25,820 dDMRs, 2.2% overlapped LMRs and 2.4% UMRs. This overlap was slightly greater than expected by chance (Bonferroni p<0.05, LMR $\log_2$FE = 1.0 and promoter-like UMRs $\log_2$FE = 1.1, *Figure 2B*) but more modest than seen for the other regulatory annotations. Similarly, we also observed that dDMRs showed more modest ($\log_2$FE = 0.4–1.0), but still significant (Bonferroni p<0.05) levels of enrichment with respect to all other islet regulatory annotations (*Figure 2B*). The modest enrichment of dDMRs indicates that only a fraction of these regions correspond to islet genomic regulatory sites. Given that T2D risk variants preferentially map in islet regulatory sites, the corollary is that most dDMRs are unlikely to directly contribute to the mediation of genetic T2D risk.

## Refining islet enhancer function using methylation and open chromatin data

To further characterise the regulatory potential of hypomethylated regions, including LMRs and UMRs, we combined the islet WGBS methylation data with chromatin accessibility data generated from ATAC-seq assays of 17 human islet samples (from non-diabetic donors of European descent; mean read count after filtering = 130M, *Figure 2—figure supplement 1C*). We identified a total of 141 k open chromatin regions based on read depth, peak width and signal-to-noise ratio (see Materials and methods). These regions of islet open chromatin showed substantial overlap (78%) with equivalent regions described in a recent study of two human islets (*Varshney et al., 2017*) ($\log_2$FE = 2.8 compared to random sites, not shown). In addition, our islet ATAC-Seq sites demonstrated substantial overlap with LMRs: 53% of LMRs overlapped 16% of all ATAC-seq peaks (LMR $\log_2$FE = 3.8 compared to randomised sites, *Figure 2B*). Almost all UMRs (98%) were contained within regions overlapping (13% of) ATAC-seq peaks (UMR $\log_2$FE = 3.4 compared to randomised sites, *Figure 2B*).

To fully leverage information across multiple overlapping islet epigenome assays, we generated augmented chromatin state maps, using chromHMM (*Ernst and Kellis, 2012*). These maps combined the WGBS methylation and ATAC-Seq open chromatin data with previously generated ChIP-seq marks (*Figure 3A*, *Figure 3—figure supplement 1A*). For these analyses, we initially used a single definition for hypomethylated regions (methylation <60%) that captured both UMRs and LMRs (see Materials and methods).

This augmented and larger set of 15 islet chromatin states retained the broad classification of regulatory elements that included promoters (positive for H3K4me3), transcribed and genic regions (H3K36me3), strong enhancers (H3K4me1; H3K27ac), weak enhancers (H3K4me1), insulators (CTCF) and repressed elements (H3K27me) (*Figure 3A*). The addition of islet methylation and open chromatin data expanded existing chromatin state definitions to provide new subclasses, particularly amongst enhancer elements. Here, we observed two subclasses of strong enhancers and three of weak enhancers (*Figure 3A*). We denote the strong enhancer subtypes as 'open' (n = 32 k genome-wide), characterised by open chromatin and hypomethylation, and 'closed' (n = 110 k) with closed chromatin and hypermethylation (*Figure 3A*). The three weak enhancer states we denote as 'open'

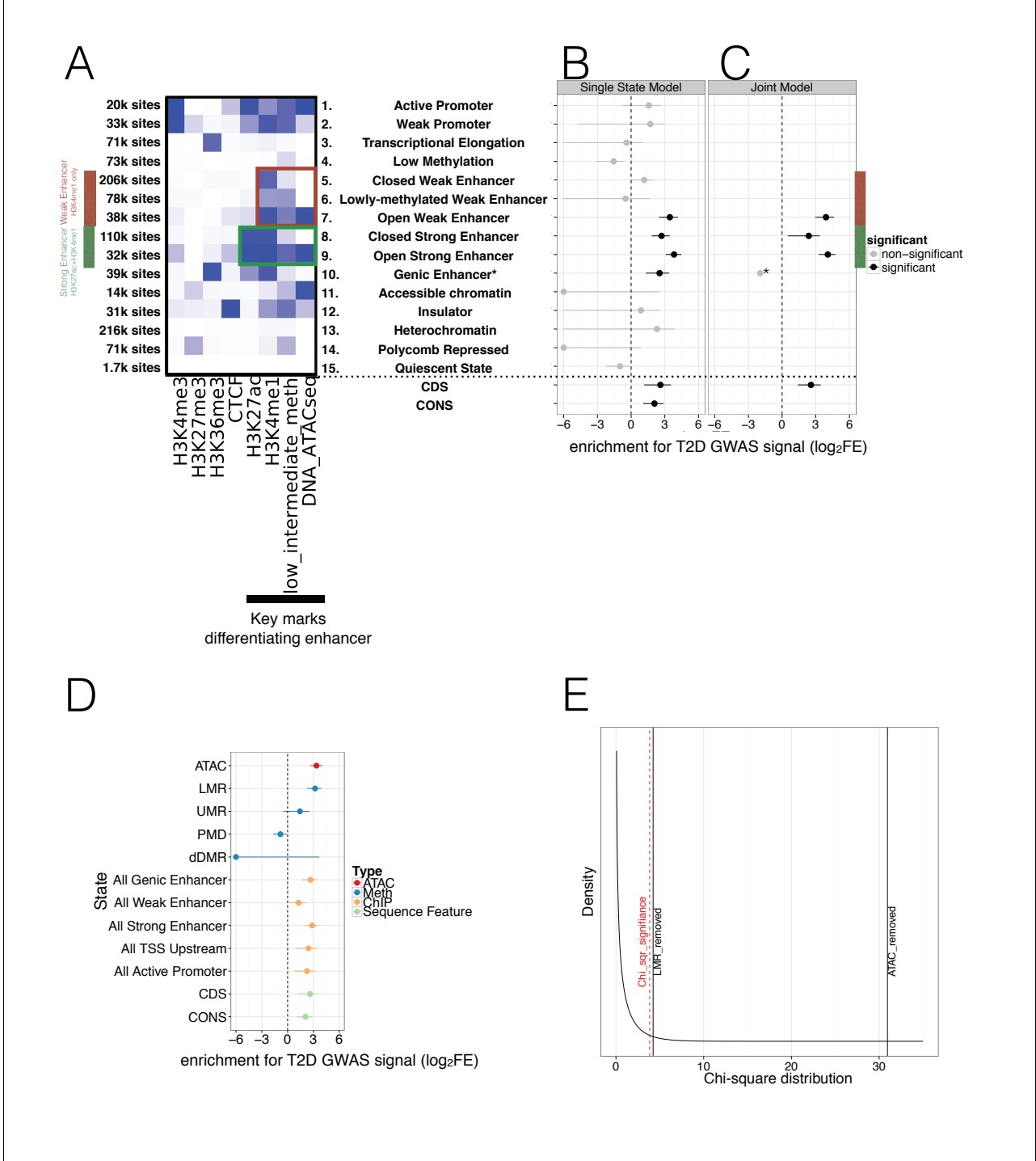

**Figure 3.** Integration of islet epigenetic data to refine chromatin regulatory states and enrichment of these states in T2D GWAS data. (A) 15 chromatin states (y-axis) were derived from ChIP histone marks, DNA methylation and ATAC-seq open chromatin annotations (x-axis) using chromHMM. For each state the relevant marks characterising the state are shown. The colour is based on the chromHMM emission parameters and a darker colour indicates a higher frequency of a mark at a given state. Weak enhancers (marked by H3K4me1 alone, red) and strong enhancers (marked by H3K27ac and H3K4me1, green) were subdivided by the chromHMM analysis according to methylation and ATAC-seq status (highlighted in red and green box). The black bar at the x-axis highlights the most important marks for characterising enhancer subtypes. (B-C) FGWAS Log$_2$ Fold Enrichment including 95% CI (log$_2$FE, x-axis) of all chromatin states (y-axis) in T2D GWAS regions is shown which demonstrate differential enrichment amongst enhancer subclasses in single-feature enrichment analysis. In addition, log2FE of Coding Sequence (CDS) and Conserved Sequence (CONS) annotations are shown to include the effect of protein-coding and conserved regions. Significantly enriched annotations are shown in black while non-siginificant annotations are

*Figure 3 continued on next page*

*Figure 3 continued*

shown in grey. (C) T2D FGWAS maximum likelihood model determined through cross-validation. Log$_2$FE and 95% CI (x-axis) of annotations included in the maximum likelihood model (y-axis) also demonstrate differential enrichment amongst enhancer subclasses. *Analysis for Genic Enhancers (state 10) did not converge and hence, only a point log2FE estimate is provided. (D) Single feature log2FE including 95% CI (x-axis) results are shown highlighting the differences in T2D GWAS enrichment of various annotations. These include ATAC-seq open chromatin peaks (red), WGBS methylation regions (including enhancer-like LMRs, promoter-like UMRs and Partially Methylated Domains, blue), ChIP-seq chromatin states (orange) and CDS and CONS (green) annotations. (E) Chi-square distribution (curved black line) with the indicated results of a maximum likelihood ratio test based on the maximum likelihood difference between a model including LMRs or ATAC-seq peaks compared to the ChIP-only model. The dashed red line indicates significance (p-value<0.05). For all FGWAS enrichment plots the axis has been truncated at −6 to facilitate visualisation and accurate values are provided in the supplementary tables.

DOI: https://doi.org/10.7554/eLife.31977.008

The following source data and figure supplements are available for figure 3:

**Source data 1.** Annotation enrichment in T2D GWAS data.
DOI: https://doi.org/10.7554/eLife.31977.011
**Source data 2.** Evaluating enrichment in T2D GWAS data.
DOI: https://doi.org/10.7554/eLife.31977.012
**Source data 3.** Evaluating enrichment in FG GWAS data.
DOI: https://doi.org/10.7554/eLife.31977.013
**Source data 4.** Merged_ATAC_seq_peaks_MThurner_Oct_2017.tds is associated with primary *Figure 3*
DOI: https://doi.org/10.7554/eLife.31977.014
**Source data 5.** Pancreatic_islet_15_chromatin_states_MThurner_Oct_2017.tds.zip is associated with primary *Figure 3*.
DOI: https://doi.org/10.7554/eLife.31977.015
**Figure supplement 1.** Prediction of regulatory regions using WGBS and ATAC-seq data and testing these regions for enrichment in T2D GWAS regions.
DOI: https://doi.org/10.7554/eLife.31977.009
**Figure supplement 2.** Enrichment of refined islet regulatory states in FG GWAS data.
DOI: https://doi.org/10.7554/eLife.31977.010

(n = 38k: open chromatin, hypomethylation), 'lowly-methylated' (n = 78 k; closed chromatin, hypo-methylation) and 'closed' (n = 206k: closed chromatin, hypermethylation). No equivalent class of 'lowly-methylated' strong enhancers was observed in the 15-state model. When comparing these chromatin states to those identified using only ChIP-seq marks ([*Parker et al., 2013*], *Figure 1—figure supplement 1C*), the two strong enhancer subclasses we identified subdivided the 'strong enhancer 1' state as described by Parker (defined by H3K27ac and H3K4me1). Additional comparison to 'stretch' enhancer clusters (*Parker et al., 2013*), showed that there was considerable overlap between the 'open' strong and weak enhancer states we identify here and previously-described 'stretch' enhancer states (16.1 k out of 23 k stretch enhancer overlapped 32 k out of 70.1 k 'open' enhancers). Even so, most (55%) 'open' enhancer states, and in particular 'open weak enhancers' (70%), were not captured within 'stretch' enhancer intervals, and we regard these as distinct islet enhancer subclasses.

To understand the relationship of these various state definitions to genetic variants influencing T2D risk, we applied the hierarchical modelling approach FGWAS (*Pickrell, 2014*) to the same sets of large-scale GWAS data for T2D (from DIAGRAM [*Scott et al., 2017*]) and FG (ENGAGE [*Horikoshi et al., 2015*]) described in section 2.1. FGWAS allowed us to combine GWAS and genomic data to determine the genome-wide enrichment within islet regulatory features for variants associated with T2D risk. These enrichment priors were then used to generate credible variant sets that are informed by both GWAS and genomic data, as described in section 2.4.

In single-feature analyses, we found significant enrichment (lower limit of Confidence Interval (CI) >0) limited to four enhancer states (open weak enhancers, both types of strong enhancer and H3K36me3 marked genic enhancers) (*Figure 3B*, *Table 1*). To take into account protein-coding variant and conserved sequence effects, we also included CoDing exon Sequence (CDS) (*Carlson and Maintainer, 2015*) and CONServed sequence (CONS) (*Lindblad-Toh et al., 2011*) as additional annotations which were previously found to be strongly enriched for T2D GWAS signal (*Finucane et al., 2015*). We observed significant enrichment for CDS and CONS sequence in the single state results (*Figure 3B*, *Table 1*). FGWAS multi-feature analyses for T2D, incorporating all annotations positive in single-element analyses, retained both subclasses of strong enhancer, the subclass

**Table 1.** Single FGWAS annotation enrichment in T2D and FG GWAS data.

For each chromatin state annotation the total number of sites and the single state FGWAS log2 Fold Enrichment (log2FE) in T2D and FG is shown. In addition, log2FE enrichment is also shown for CDS and CONS annotation. 95% Confidence Intervals (CI) for log2FE are shown in brackets and significantly enriched states are highlighted in bold (lower CI limit >0). Parker enhancer states refer to enhancer states defined by *Parker et al., 2013*.

| Chromatin States | Total number of states | T2D log2FE (CI) | FG log2FE (CI) |
|---|---|---|---|
| 1. Active Promoter | 20 k | 1.6 (-0.8 to 2.7) | 2.7 (0 to 4.1) |
| 2. Weak Promoter | 33 k | 1.7 (-4.8 to 2.9) | 2.7 (-0.1 to 4.2) |
| 3.Transcriptional Elongation | 71 k | -0.4 (-20 to 1.1) | -26.1 (-46.1 to 1.0) |
| 4. Low Methylation | 73 k | -1.5 (-3.1 to -0.6) | -1.7 (-4.2 to -0.3) |
| 5. Closed Weak Enhancer | 206 k | 1.2 (-0.1 to 2) | 1.7 (0 to 2.9) |
| 6. Lowly-methylated Weak Enhancer | 78 k | -0.5 (-20 to 1.6) | -26.7 (-46.7 to 1.6) |
| 7. Open Weak Enhancer | 38 k | **3.4 (2.5 to 4.2)** | 3.1 (-0.6 to 4.6) |
| 8. Closed Strong Enhancer | 110 k | **2.7 (1.8 to 3.4)** | **3.3 (2 to 4.4)** |
| 9. Open Strong Enhancer | 32 k | **3.8 (3.1 to 4.5)** | **4.3 (2.8 to 5.5)** |
| 10. Genic Enhancer | 39 k | **2.5 (1.3 to 3.4)** | **2.9 (0.8 to 4.3)** |
| 11. Accessible chromatin | 14 k | -25.2 (-45.2 to 2.5) | -28.4 (-48.4 to 3.7) |
| 12. Insulator | 31 k | 0.9 (-20 to 2.6) | -0.6 (-20 to 3.6) |
| 13. Heterochromatin | 216 k | 2.3 (-20 to 3.9) | 1.8 (-1.5 to 4.0) |
| 14. Polycomb Repressed | 71 k | -25.5 (-45.5 to 0.9) | -33.2 (-53.2 to 1.5) |
| 15. Quiescent State | 1.7 k | -1.0 (-2.2 to -0.1) | -28.6 (-48.6 to -0.6) |
| CDS | NA | **2.6 (1.2 to 3.5)** | 2.7 (-0.2 to 4.3) |
| CONS | NA | **2.1 (1.1 to 2.9)** | **1.9 (0.2 to 3.2)** |
| Parker Weak Enhancer | 119 k | 0.9 (-2.5 to 2.0) | -2.0 (-20.0 to 2.4) |
| Parker Strong Enhancer (all) | 123 k | **2.7 (2.0 to 3.3)** | **3.1 (2.0 to 4.4)** |
| Parker Strong Enhancer (open) | 64 k | **3.1 (2.4 to 3.7)** | **3.6 (2.3 to 4.8)** |
| Parker Strong Enhancer (closed) | 59 k | **1.9 (0.8 to 2.7)** | **2.3 (0.5 to 3.5)** |

DOI: https://doi.org/10.7554/eLife.31977.016

of open weak enhancers, genic enhancers and CDS in the joint model (*Figure 3C* and Materials and methods). Conserved sequence annotations were not retained in the joint model.

We observed markedly different levels of enrichment for T2D association between and within open and closed enhancer states (*Figure 3B–3C*, *Table 1*). Using these augmented chromatin state maps, we demonstrated clear enrichment for T2D association for the subset of 'open' weak enhancers (12% of all weak enhancer sites) with no evidence of enrichment in the remaining sub-classes ('closed' and 'lowly-methylated') (*Figure 3B* and *Table 1*). This concentration of enrichment amongst a relatively small subset of the weak enhancers was consistent with the lack of enrichment across all weak enhancers defined solely on the basis of H3K4me1 signal ([*Parker et al., 2013*], single state $\log_2$FE = 0.9, CI = −2.5 to 2.0, *Table 1*, *Figure 1—figure supplement 1C*). We also saw differences in enrichment signal between open and closed strong enhancers, with the most marked enrichment amongst open strong enhancers (22% of the total, *Figure 3B–C*, *Table 1*). This effect was particularly obvious in the joint-analysis (open strong enhancer joint $\log_2$FE = 4.1, CI = 3.3 to 4.8 vs. closed strong enhancer joint $\log_2$FE = 2.4, CI = 0.5 to 3.3, *Figure 3C*).

Hypomethylation and open chromatin are highly correlated, but the observed difference in T2D enrichment between the weak enhancer states (particularly between 'lowly-methylated' and 'open' which differ markedly with respect to chromatin status) points to a primary role for open chromatin. To test this further, we regenerated chromatin state maps using different subsets of the data (ChIP-only, with optional addition of methylation and/or open chromatin information, see Materials and methods and *Figure 3-figure supplement 1A-B*). These analyses confirmed that the T2D GWAS

enrichment signal was predominantly driven by the distribution of islet open chromatin (*Figure 3—figure supplement 1C*).

We further evaluated the role of subclasses of DNA methylation regulatory region with respect to T2D GWAS enrichment. We divided hypomethylated (<60% methylated) sequence into enhancer-like LMRs (6.5% of all hypomethylated sequence), promoter-like UMRs (7.5% of hypomethylated sequence), as well as PMDs (61% of hypomethylated sequence). The remaining 25% of hypomethylated sequence did not fit any category. LMRs were significantly (lower CI limit >0) enriched ($\log_2$FE = 3.2, CI = 2.3 to 3.9) for T2D association signals consistent with their co-localisation with distal regulatory elements, and displayed modestly increased enrichment compared to enhancer states derived from ChIP-seq alone (*Figure 3D*, *Figure 3—source data 1*). In contrast, no significant enrichment was found for human islet (promoter-like) UMRs ($\log_2$FE = 1.4, CI = −0.6 to 2.5) or PMDs ($\log_2$FE = −0.8, CI = −1.7 to −0.1). We also found no evidence that recently-described regions of T2D-associated differential methylation (dDMRs: derived from comparison of WGBS data from islets of diabetic and non-diabetic individuals) were enriched for genome-wide T2D association signals ($\log_2$FE = −24.6, CI = −44.6 to 3.7) (*Figure 3D*, *Figure 3—source data 1*).

Finally, since the hypomethylation signal for T2D enrichment was concentrated in LMRs (*Figure 3D*, *Figure 3—source data 1*), we reran a FGWAS joint-analysis combining open chromatin peaks, LMRs and ChIP-only states using a nested model (*Figure 3E*, *Figure 3—figure supplement 1D–E*, see Materials and methods). This confirmed that the improvement in enrichment was mainly driven by open chromatin but showed that LMRs also contributed significantly and independently to the enrichment (*Figure 3E*, *Figure 3—source data 2*).

FGWAS analysis for FG corroborated the observations from T2D analysis. Despite reduced power of the FG GWAS data due to a lower number of significantly associated FG GWAS loci, both single feature and joint-model analyses of human islet epigenome data found significant enrichment in strong enhancer states with the strongest enrichment in enhancers with open chromatin and hypomethylation (*Figure 3—figure supplement 2A–B* and *Table 1*). In addition, evaluation of the relative contributions of ATAC-seq open chromatin and DNA methylation to FG GWAS enrichment across both single-feature (*Figure 3—figure supplement 2C–D*) and joint-model analysis (*Figure 3—figure supplement 2E–F* and *Figure 3—source data 3*) indicated that open chromatin was primarily responsible for the enhanced enrichment.

Overall, these analyses demonstrate that the addition of open chromatin and DNA methylation data to ChIP-seq marks enhances the resolution of regulatory annotation for human islets. In particular, it defines subsets of weak and strong enhancers that differ markedly with respect to the impact of genetic variation on T2D risk. Although DNA accessibility and hypomethylation status are strongly correlated and provide broadly similar enrichments, the effects of the former predominate. In line with the dominance of open chromatin status for T2D GWAS enrichment, we observed that T2D risk in relation to methylation status is primarily invested in hypomethylated LMRs (i.e. enhancers) rather than UMRs, dDMRs or PMDs.

## Augmented chromatin maps and open chromatin allelic imbalance refine likely causal variants at *ADCY5*, *CDC123*, and *KLHDC5*

We next deployed the insights from the global FGWAS enrichment analyses to define the molecular mechanisms at individual T2D susceptibility loci, refining T2D causal variant localisation using the combination of genetic data (from fine-mapping) and the genome-wide patterns of epigenomic enrichment.

Specifically, we applied FGWAS to the T2D DIAGRAM GWAS data (*Scott et al., 2017*) under the joint model (*Figure 3C*) derived from the augmented chromatin state maps. We divided the genome into 2327 segments (average size 5004 SNPs or 1.2 Mb) and identified 52 segments significantly associated with T2D genome-wide (segmental FGWAS PPA >= 0.9 or single variant GWAS $p < 5 \times 10^{-8}$, see Materials and methods for details). These corresponded to 49 known T2D associated regions representing that subset of the ~120 known T2D GWAS loci which passed those significance/filtering criteria in this European-only dataset. We then calculated reweighted PPAs for each variant within each segment and generated reweighted 99% credible sets. (Of note, in line with traditional GWAS nomenclature, locus names were defined based on proximity between the lead variant and the closest gene and does not, of itself, indicate any causal role for the gene in T2D susceptibility).

Consistent with the increased T2D GWAS enrichment of states including open chromatin and DNA methylation information, we found that analyses using enrichments from the augmented chromatin state model (combining ChIP-seq, ATAC-seq and WGBS data) were associated with smaller 99% credible sets (median of 17 SNPs) than those derived from FGWAS enrichment derived from ChIP-seq data alone (median 23). In parallel, the PPA for the best variant per locus increased (median 0.39 vs 0.31). Individual T2D GWAS locus results are shown in *Figure 4A–B*. We also expanded the FGWAS PPA analysis to investigate open chromatin and DNA methylation effects on fine-mapping and found that the reduction in 99% credible set size and increase in maximum variant PPA was driven predominantly by open chromatin (*Figure 4—figure supplement 1*, *Figure 4—source data 1*). This demonstrates that the inclusion of open chromatin maps helps to improve prioritisation of causal variants at many T2D GWAS loci.

A subset of T2D GWAS signals are known to influence T2D risk through a primary effect on insulin secretion, whilst others act primarily through insulin resistance. We used previous categorisations of T2D GWAS loci based on the patterns of association with quantitative measurements of metabolic function and anthropometry (*Wood et al., 2017*; *Dimas et al., 2014*), to define a set of 15/48 loci most clearly associated with deficient insulin secretion (and therefore most likely to involve islet dysfunction). At 11 of these 15 loci, we found that islet 'open strong enhancer' states, and to a lesser extent 'open weak enhancer' and 'closed strong enhancer', captured more than 60% of the PPA (median 92%, *Figure 4C*). Variants in these islet enhancer subclasses also captured at least 95% of the PPA at 4 T2D GWAS loci that could not be classified according to physiological association data but which have been previously implicated in human islet genome or functional regulation based on islet eQTL (*van de Bunt et al., 2015*) or mQTL (*Olsson et al., 2014*) data (*Figure 4C*, genes highlighted in bold). In contrast, at 3/6 of the insulin resistance and all but five unclassified loci, the PPA was mostly (>50%) attributable to other non-islet enhancer states (across all insulin resistance and unclassified loci, DNA not overlapping islet enhancers and defined as 'Other' capture a median PPA of 64%). Thus, islet regulatory annotations are particularly useful for fine-mapping T2D GWAS loci that affect insulin secretion and beta-cell function.

To obtain additional evidence to support the localisation of causal variants, we tested for allelic imbalance in ATAC-seq open chromatin data. We selected 54 variants within 33 T2D-associated GWAS segments for testing of allelic imbalance on the basis of (a) a reweighted variant PPA >= 10% and (b) overlap with an enriched regulatory state within the FGWAS T2D joint-model (*Figure 4D*, *Figure 4—source data 2*). Of these, 20 variants (at 16 loci) had sufficient numbers of heterozygous samples (>2) and ATAC-seq read depth (depth >9 and at least five reads for each allele). After correcting for mapping bias using WASP, we observed the strongest evidence for allelic imbalance (FDR < 0.05) at 3 out of the 20 variants (rs11257655 near *CDC123* and *CAMK1D*, rs10842991 near *KLHDC5* and rs11708067 at *ADCY5*) (*Table 2*). All three overlapped refined islet open strong or open weak enhancer regions characterised by open chromatin and hypomethylation.

Variant rs11257655 accounts for 95% of the reweighted PPA (compared to a PPA of 20% from genetic data alone) at the *CDC123/CAMK1D* locus, overlaps an 'open strong enhancer' region (*Figure 5A*) and the risk allele correlates with increased chromatin accessibility. The same variant is in high LD ($r2 = 0.82$) with the lead variant for a cis-eQTL for *CAMK1D* in islets (*van de Bunt et al., 2015*). In experimental assays (*Fogarty et al., 2014*), the T2D-risk allele has been shown to be associated with increased *CAMK1D* gene expression and enhanced binding of the FOXA1 and FOXA2 transcription factors. These data all point to rs11257655 as the causal variant at this locus.

At *KLHDC5*, no clear causal variant emerged based on genetic fine-mapping data alone as the credible set contained 23 variants in high mutual LD ($r^2 >0.8$, top variant PPA <5%, *Figure 5B*). Of these, variants rs10771372 (genetic fine-mapping PPA = 5%), rs10842992 (genetic fine mapping PPA = 5%) and rs10842991 (genetic fine-mapping PPA = 3%) overlapped 'open strong enhancer' regions (*Figure 5B*), such that their reweighted PPAs rose to 21% (rs10771372), 21% (rs10842992) and 13% (rs10842991), respectively. We observed allelic imbalance only at rs10842991 with the T2D-risk C allele showing greater chromatin accessibility (binomial $p=4.1\times10^{-3}$, *Table 2*). This variant further overlapped a predicted TFBS motif for PAX6 as determined by the software tool FIMO (*Grant et al., 2011*): the T2D-risk allele was predicted to enhance PAX6 transcription factor binding consistent with the allelic effects on increasing chromatin accessibility (*Figure 5—figure supplement 1A*). This strong enhancer region is almost exclusively found in islets, with strong enhancer H3K27ac states overlapping rs10842991 in only two non-islet (heart and smooth muscle) Epigenome Roadmap

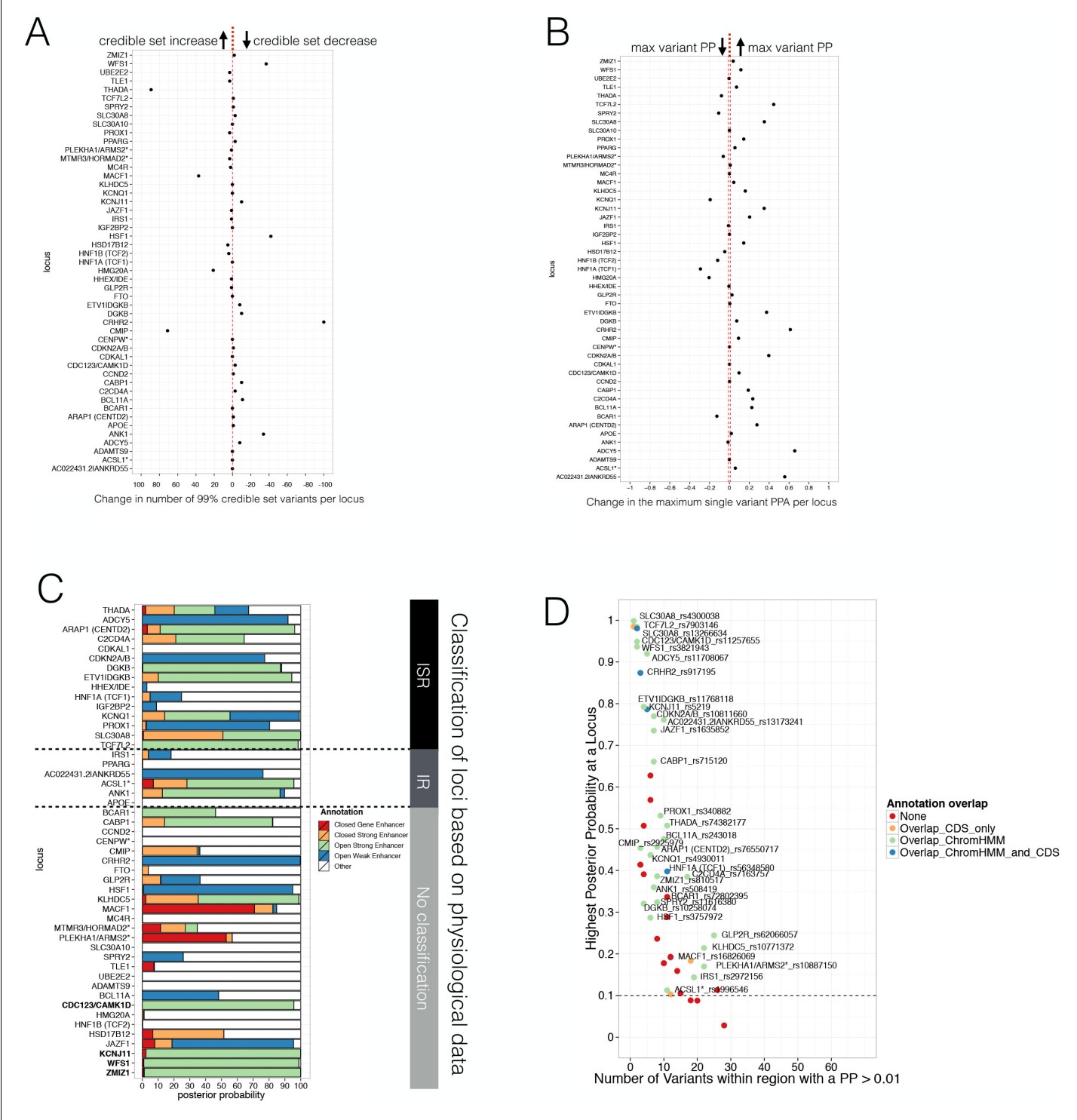

**Figure 4.** Evaluating Posterior Probabilities (PPA) derived from the FGWAS maximum likelihood model at significant T2D GWAS loci. (**A**) Per locus the difference in the number of 99% credible set variants between ChIP +ATAC + Meth and ChIP-only model is shown (positive values indicate a reduction in the number of 99% credible set variants in the ChIP+ATAC+Meth model). (**B**) Per locus the difference in the maximum single variant PPA between the ChIP +ATAC + Meth and ChIP-only model is shown (positive values indicate an increase in the maximum single variant PPA in the ChIP +ATAC + Meth model). (**C**) T2D GWAS loci were classified into insulin secretion (ISR), insulin resistance (IR) or unclassified loci based on genetic association with physiological traits derived from *Dimas et al. (2014)* and *Wood et al. (2017)*. In addition, loci with known role in islet genomic regulation or function are highlighted in bold. These include loci with islet eQTLs (*ZMIZ1, CDC123*) and mQTLs (*WFS1, KCNJ11*). (**D**) Identification of T2D GWAS loci and variants enriched for enhancer chromatin states using FGWAS PPA. Per locus the highest PPA variant is shown (y-axis) and the number of variants with

*Figure 4 continued*

PPA >0.01 (x-axis). Loci with high PPA variants (min PPA >0.1, dashed horizontal line) that overlap one of the enhancer states (green) are highlighted and the high PPA variants (PPA >0.1) were tested for allelic imbalance in open chromatin.

DOI: https://doi.org/10.7554/eLife.31977.017

The following source data and figure supplement are available for figure 4:

**Source data 1.** Comparison of variant variant PPA and 99% credible set size across annotations.
DOI: https://doi.org/10.7554/eLife.31977.019

**Source data 2.** Information for variants overlapping a genomic annotation included in the FGWAS T2D-joint model.
DOI: https://doi.org/10.7554/eLife.31977.020

**Figure supplement 1.** Evaluating annotation effect on Posterior Probabilities (PPA) derived from the FGWAS maximum likelihood model at significant T2D GWAS loci.
DOI: https://doi.org/10.7554/eLife.31977.018

tissues (out of 99 tissues with 18-state chromatin state information, *Figure 5B*)(*Kundaje et al., 2015*). Islet eQTL data (*Varshney et al., 2017*) also links rs10842991 and close proxy SNPs (including rs7960190) to islet transcription with the risk allele increasing *KLHDC5* expression. These data prioritise rs10842991 as the likely causal variant at the *KLHDC5* T2D GWAS locus, and indicate a likely molecular mechanism involving modified PAX6 transcription factor binding and an impact on *KLHDC5* expression and islet function.

The third example of allelic imbalance mapped to the *ADCY5* locus. Fine-mapping based solely on genetic data could not prioritise a distinct causal variant due to multiple variants in high LD (range for top five variants = 12–26%, *Figure 5C*). However, reweighting of variants based on epigenomic annotation clearly prioritised variant rs11708067: this SNP overlapped an 'open weak enhancer' and captured most of the reweighted PPA (PPA = 92%). Allelic imbalance analysis also showed that the T2D-risk A allele was associated with decreased chromatin accessibility (binomial p=$1.2\times10^{-6}$, *Table 2*). The same lead variant maps to an islet cis-eQTL and methylation QTL (*Figure 5C*, *Figure 5—figure supplement 1B*) at which the T2D-risk allele is associated with reduced *ADCY5* expression and increased *ADCY5* gene body DNA methylation.

To further understand the role of the rs11708067 variant, we performed ATAC-seq and Next Generation Capture-C, in the glucose-responsive human beta-cell line EndoC-βH1 (n = 3). We targeted the *ADCY5* promoter to define distal regions interacting with the promoter, and confirmed physical contact with the hypomethylated open chromatin enhancer region harbouring rs11708067 (*Figure 5C*, *Figure 5—figure supplement 1C*). To resolve the significance of the interaction between the restriction fragment encompassing rs11708067 and the *ADCY5* promoter, we used the programme peakC (*de Wit and Geeven, 2017*) (https://github.com/deWitLab/peakC) to evaluate the interactions of 12 fragments covering the lead SNP rs11708067 and 15 SNPs in high LD (r2 >0.8) across a region of 47 kb. After adjusting for multiple testing using FDR correction, only two fragments yielded a significant normalised read number over background. This included the open-chromatin overlapping fragment containing rs11708067 and another fragment harbouring rs2877716, rs6798189, rs56371916 (*Figure 5—figure supplement 1D*). These SNPs fall into a region that did not show evidence of open chromatin.

**Table 2.** T2D-associated variants with allelic imbalance in open chromatin.

| Variant | Locus | DIAGRAM P-value | Fgwas T2D PPA | Allelic imbalance Allele Ratio (Allele #) | Allelic imbalance WASP P-value | Direction of effect (T2D) |
|---|---|---|---|---|---|---|
| rs11708067 | ADCY5 | 8.8E-13 | 0.92 | 0.29 (38 A VS 94 G alleles) | 1.2E-06 | risk allele A closed |
| rs11257655 | CDC123 | 4.0E-08 | 0.95 | 0.39 (278 C VS 435 T alleles) | 4.5E-09 | risk allele T open |
| rs10842991 | KLHDC5 | 7.3E-07 | 0.13 | 0.64 (75 C VS 43 T alleles) | 4.1E-03 | risk C allele open |

DOI: https://doi.org/10.7554/eLife.31977.021

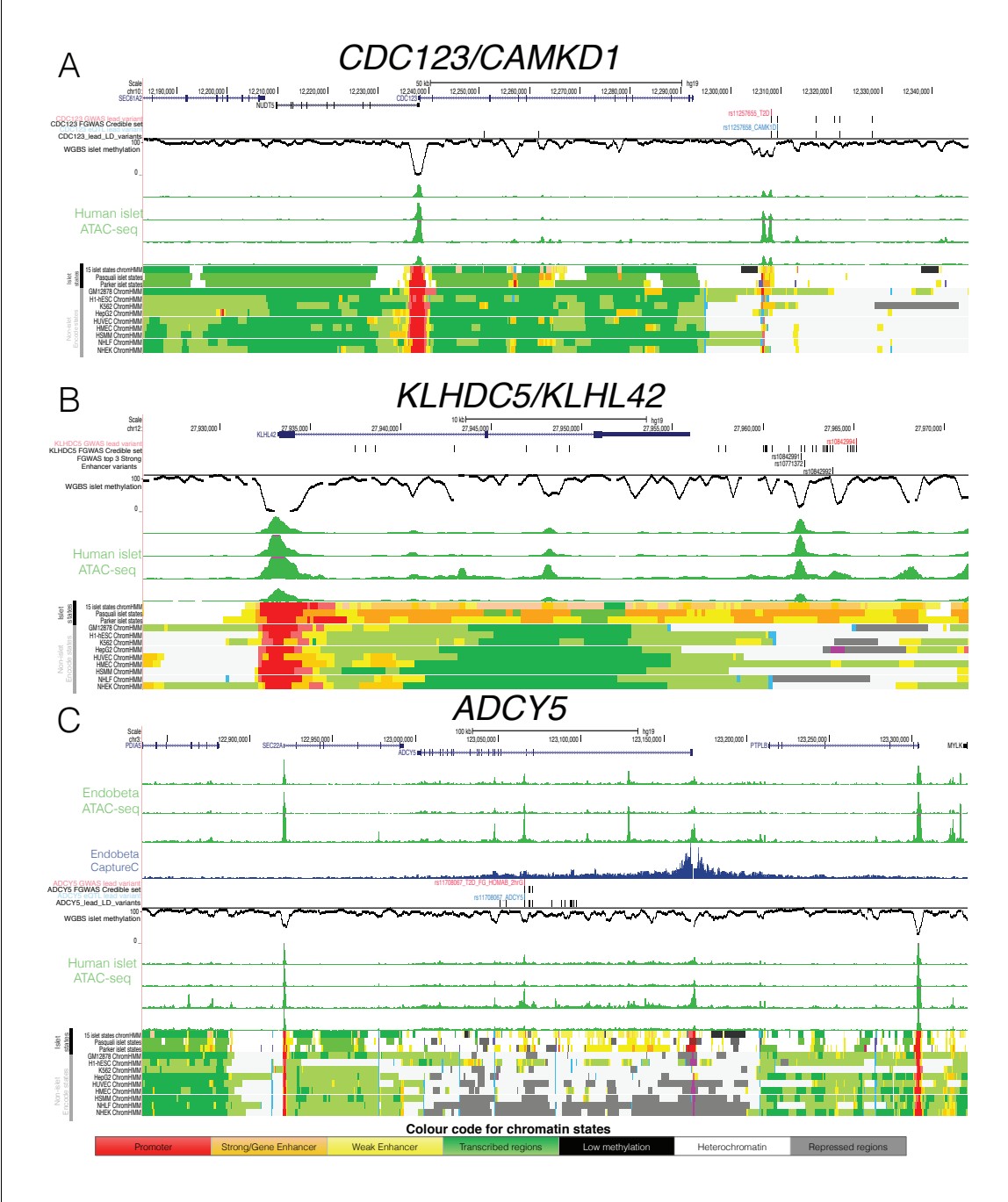

**Figure 5.** Epigenome Landscape of selected loci with allelic imbalance. For each locus (**A**) *CDC123*, (**B**) *KLHDC5* and C) *ADCY5* the following information is shown: Variant level information (depending on the region GWAS lead SNP red, credible set black, eQTL blue and high LD SNPs with r2 >0.8 black), WGBS methylation data (black, middle), four human islet ATAC-seq tracks (green, middle), islet chromatin states (from this study as well as *Parker et al., 2013*) and *Pasquali et al., 2014*) and Encode chromatin states from 9 cell types (bottom). For *ADCY5* 3 ATAC-seq Endoß tracks (top green) and the Capture C results in the Endoß cell line are shown as well (middle blue). Abbreviation for cell types: B-lymphoblastoid cells (GM12878), embryonic stem cells (H1 ES), erythrocytic leukaemia cells (K562), hepatocellular carcinoma cells (HepG2), umbilical vein endothelial cells (HUVEC), mammary epithelial cells (HMEC), skeletal muscle myoblasts (HSMM),normal epidermal keratinocytes (NHEK) and normal lung fibroblasts (NHLF).
DOI: https://doi.org/10.7554/eLife.31977.022

The following figure supplement is available for figure 5:

**Figure supplement 1.** Characterisation of likely causal mechanisms at selected loci with allelic imbalance.
DOI: https://doi.org/10.7554/eLife.31977.023

These findings support rs11708067 as the likely causal variant affecting islet accessible chromatin (in line with another recent study [*Roman et al., 2017*]), and link the open and hypomethylated enhancer element in which it sits to regulation of *ADCY5* expression in islets.

## Discussion

A key challenge in the quest to describe the molecular mechanisms through which GWAS signals influence traits of interest, involves the identification of the causal variants responsible and, given that most lie in non-coding sequence, the characterisation of the regulatory elements which they perturb. This underpins efforts to define the effector genes through which these variants operate and to reconstruct the biological networks that are central to disease pathogenesis.

Genetic and physiological studies have highlighted the singular importance of pancreatic islet dysfunction in type 2 diabetes, but epigenomic characterisation of this tissue has been limited in large-scale community projects such as ENCODE and GTEx. The present study seeks to address this deficit by describing, in unprecedented detail, genome-wide patterns of methylation and chromatin accessibility in human islet material. We have combined these data with existing islet epigenomic marks to generate a refined regulatory map which, based on the evidence of improved enrichment for T2D association signals, offers more granular annotation of functional impact.

Our data show that, for DNA methylation, the signal of T2D predisposition is primarily associated with enhancer-like LMRs rather than other categories of methylation elements including UMRs, dDMRs or PMDs. We highlight the strong correlation between islet methylation status and chromatin accessibility but demonstrate that open chromatin predominantly contributes to defining the regulatory impact associated with genetic T2D risk. Finally, we demonstrate how these enhanced epigenomic annotations, when analysed in concert with genetic fine-mapping data and information from allelic imbalance in chromatin accessibility allow us to home in on likely causal variants at T2D association signals such as those near *ADCY5, CDC123* and *KLHDC5*.

While previous studies had explored the candidacy of selected variants at the *CDC123* (*Fogarty et al., 2014*) and *ADCY5* (*Olsson et al., 2014*; *Hodson et al., 2014*; *van de Bunt et al., 2015*) loci with respect to islet regulation and T2D predisposition, our integrative analysis of T2D GWAS and epigenetic data has enabled a detailed and comprehensive analysis that considers the regulatory impact of all variants at these loci across multiple islet samples. Our analysis implicates the rs11257655 and rs11708067 variants as the most likely causal variants at the *CDC123* and *ADCY5* loci respectively and highlights their relationship to islet enhancer activity. The findings at *ADCY5* are supported by a recent paper that found allelic imbalance in H3K27 acetylation involving the rs11708067 variant in a single human islet sample, and which observed that deletion of the relevant enhancer element led to reduction in both *ADCY5* gene expression and insulin secretion (*Roman et al., 2017*).

At the *KLHDC5* locus, local LD frustrated efforts to define the causal variant using genetic data alone, but the integration of genetic and epigenetic data pinpointed rs10842991 as the likely culprit based on its impact on chromatin accessibility in an open enhancer region. Evidence that this variant co-localises with an islet cis-eQTL signal points to *KLHDC5* as the likely downstream target (*Varshney et al., 2017*). Overall, our integrative approach provides useful insights into the functional mechanisms through which T2D GWAS signals operate. Our findings mirror those from other studies, which have, in various ways, and for other complex traits, combined diverse epigenomic annotations to explore the basis of genetic risk (*Wang et al., 2016*).

The whole genome methylation data generated in the present study also allowed us to evaluate the likely contribution of previously identified T2D-associated dDMRs (*Volkov et al., 2017*) with respect to T2D predisposition. These dDMRs, defined on the basis of observed differences in methylation between islets recovered from diabetic and non-diabetic donors, cover a substantial part of the genome, but we were able to show that only a small minority of these overlap functional islet regulatory regions. As a consequence, dDMR regions as a whole had no significant enrichment for T2D association signals. This suggests that most of the dDMR signal involves stochastic effects and/or the secondary consequences on methylation of the diabetic state. However, we cannot exclude that some of the dDMR signals are causal contributors to the diabetic phenotype either because they reflect environmental rather than genetic predisposition, or because they accelerate further perturbation of islet dysfunction as diabetes develops.

Although we provide highly detailed functional fine-mapping of T2D genetic variants to uncover causal variants, the FGWAS approach applied in this study is limited in its ability to determine the effect of multiple variants at individual loci. Specifically, FGWAS relies on the assumption of a single causal variant within each region, which may not necessarily be true for all loci. This assumption could be violated where there are multiple independent signals at a given locus, or where there are multiple (small effect size) variants on a single risk haplotype which jointly impact the phenotype. Analysis methods that combine functional fine-mapping with conditional analysis and consider LD and haplotype patterns are likely to provide a more complete overview of the causal interactions at T2D GWAS loci.

In addition, while the present study characterises islet epigenome status and variability in chromatin accessibility in substantially larger numbers of islet samples than those previously reported (*Gaulton et al., 2015*; *Parker et al., 2013*; *Pasquali et al., 2014*; *Varshney et al., 2017*), the number of islet preparations for which these data were available was still limited. As a result, our power to detect allelic imbalance in chromatin accessibility was restricted to sites with common variants and relatively large effects. We anticipate that expansion of these sample numbers will extend our capacity to detect such allelic imbalance, and offer more granular insights into the relationships between genetic variation and methylation status. A further limitation is that the genomic data we analysed was generated only from islet samples from non-diabetic donors. Whilst causal inference is possible through the integration of basal epigenomic annotations with genetic data, addition of epigenomic data from islets recovered from diabetic donors has the potential to add a further dimension to such analyses, and to unravel what are likely to be complex causal relationships between genetic variants, epigenomic phenotypes and disease states (*Gutierrez-Arcelus et al., 2013*). Finally, future work should also focus on experimental validation of likely causal variants and mechanisms e.g. differential binding of the TF PAX6 could be tested at the *KLHDC5* rs10842991 variant through electrophoretic mobility shift assays. Our ongoing research efforts are now concentrated on improving the fine-mapping analysis and expanding these genomic enrichment analyses in larger numbers of human islet samples from healthy and diabetic islets. By coupling the integration of these data with empirical functional studies, we expect to provide an increasingly complete description of the causal interactions between DNA methylation, chromatin state, RNA expression and T2D susceptibility.

## Materials and methods

### Human pancreatic islet samples

WGBS and 450 k array human pancreatic islet sample collection

Human islets were retrieved from deceased Caucasian non-diabetic donors from the Oxford DRWF Human Islet Isolation Facility (n = 34) and at the Alberta Diabetes Institute in Edmonton in Canada (n = 10). For the analysis only samples with a purity >70% were used as determined by dithizone labeling. The Human Research Ethics Board at the University of Alberta (Pro00001754), the University of Oxford's Oxford Tropical Research Ethics Committee (OxTREC Reference: 2–15), or the Oxfordshire Regional Ethics Committee B (REC reference: 09/H0605/2) approved the studies. All organ donors provided informed consent for use of pancreatic tissue in research.

For all WGBS (n = 10) and a subset of 450 k array samples (n = 18) human pancreatic islet DNA was extracted from 100,000 to 150,000 islet cells using Trizol-(Ambion from Thermo Fisher Scientific, Waltham, MA was used for islets processed in Oxford, UK while Trizol from Sigma Aldrich, St. Louis, MO was used for islets processed in Edmonton, Canada) as described previously (*van de Bunt et al., 2015*). For the remaining 23 samples islet DNA was extracted using the ReliaPrep gDNA Tissue Miniprep system (Promega, Madison, WI). Extracted DNA was stored at −80°C before further use.

ATAC-seq human pancreatic islet sample collection

Human pancreatic islets preparations (n = 18) were retrieved from 17 deceased non-diabetic donors of European descent from the Oxford DRWF Human Islet Isolation Facility and stored for 1–3 days in CMRL or UW media. The latter were reactivated in CMRL for 1 hr before processing them further. Approximately 50,000 islet cells per sample were hand-picked and immediately processed for

ATAC-seq as described previously (*Buenrostro et al., 2013*), however, an additional round of purification was performed using Agencourt AMPure XP magnetic beads (Beckman Coulter, Brea, CA).

## WGBS data generation

### Bisulphite conversion

400 ng of DNA per human islet samples (n = 10) were sent as part of a collaborative effort to the Blizard Institute, Queen Mary University, London, UK and bisulphite- converted using the Ovation Ultralow Methyl-Seq DR Multiplex System 1–8 (Nugen, Manchester, UK) and purified using Agencourt AMPure beads (Beckman Coulter) as described previously (*Lowe et al., 2013*).

### Library generation and processing of reads

The libraries were sequenced by members of the High-Throughput Genomics group at the Wellcome Centre for Human Genetics, University of Oxford, Oxford, UK. Samples were sequenced as multiplex libraries across 3 HiSeq2000 lanes with 100 bp paired-end read length (including a PhiX spike-in of 5%) to obtain high-coverage read data. The obtained reads were trimmed using a customized python3 script (10 bp at the start and 15 bp at the end) and aligned to hg19 using the software Bismark (settings: L,0,–0.6, version 0.12.5, RRID:SCR_005604)(*Krueger and Andrews, 2011*). Specifically, paired-end alignment of trimmed reads was performed and unmapped reads from read one were realigned using Bismark and merged with the paired-end alignment using samtools (*Li et al., 2009*) (version 0.1.19, RRID:SCR_002105) in order to increase mapping efficiency. Coverage for the merged paired-end and realigned HiSeq read alignments was estimated for the human mappable genome (NCBI hg19 2.8 billion base pairs excluding gaps and unmappable and blacklisted regions according to UCSC and Encode [*ENCODE Project Consortium, 2012*]) using bedtools (version v2.21.0) (*Quinlan, 2014*).

### WGBS DNA methylation quantification and prediction of hypomethylated regulatory regions

CpG methylation levels were determined for each sample by calculating the ratio of unmodified C (methylated) and bisulphite converted T (unmethylated) alleles using BiFAST (first described here [*Lowe et al., 2013*]). High-pass pooled WGBS data was generated by adding methylated and unmethylated read counts across individual low-pass samples to then estimate the average beta methylation levels.

Regulatory regions were identified using the R package methylseek (RRID:SCR_006513) (*Burger et al., 2013*). After removing PMDs, which represent highly heterogenous methylation states determined by DNA sequence features (*Gaidatzis et al., 2014*), LMRs (<30 CpGs) and UMRs (>30 CpGs) were predicted in hypomethylated regions (<50%) at an FDR of 0.05. The methylation level and FDR parameter was inferred from the data as suggested by the methylseek workflow (*Burger et al., 2013*).

## 450 k DNA methylation array data generation

In total, 41 samples were processed for the Illumina Infinium HumanMethylation450 BeadChip (Illumina, San Diego, CA). Of these 18 samples were bisulphite-converted and processed as part of a collaboration at the UCL Cancer Institute, University College London, London, UK while the remaining 23 samples were processed in OCDEM, University of Oxford, Oxford, UK. The DNA was bisulphite converted using the EZ DNA Methylation™ Kit ( Zymogen Research Corp, Irvine, CA) and hybridised to the Illumina 450 k array and scanned with iScan (Illumina) according to the manufacturer's protocol.

The resulting data was analysed using the Package minfi (RRID:SCR_012830) (*Aryee et al., 2014*) and custom R scripts ([*R Development Core Team, 2011*], R version 3.0.2, RRID:SCR_001905). Specifically, CpG sites with a detection p-value>0.01 were removed from the analysis and samples with >5% of CpG sites failing this threshold (n = 9) were also removed from the analysis.

Following separate quantile normalisation of signal intensities derived from methylated and unmethylated Type I probes and Type II probes, methylation levels (ß) were estimated, based on the intensities of the methylated (M) and unmethylated (U) signal in the following way: β = M/

(M + U + 100). To correct for batch effects the ComBat function implemented in the sva (*Johnson et al., 2007*; *Leek et al., 2007*) package was used (*Figure 1—figure supplement 2*).

## ATAC-seq data generation
### Sequencing of ATAC-seq reads
ATAC-seq libraries were sequenced at the High-Throughput Genomics group which is part of the Wellcome Centre for Human Genetics, University of Oxford, Oxford, UK. Samples were sequenced as 4-6plex libraries across 1–3 Hiseq2500 lanes with 50 bp paired-end read length.

### Processing of ATAC-seq reads
Raw FASTQ reads were processed with an in-house pipeline (first described in (*Hay et al., 2016*) and on the website http://userweb.molbiol.ox.ac.uk/public/telenius/PipeSite.html). Specifically, library and sequencing quality was checked with FASTQC (RRID:SCR_014583) (http://www.bioinformatics.babraham.ac.uk/projects/fastqc) and reads were mapped to the human genome (hg19) via bowtie (*Langmead et al., 2009*) (version 1.1.0, RRID:SCR_005476) with default settings but -m 2, and maxins 2000 which allows mapping of reads with a maximum number of 2 alignments and a maximum insert size of 2000 bp. For reads that could not be aligned the first time, adapters were removed with Trim Galore at the three prime end (RRID:SCR_011847, settings -length 10, -qualFilter 20, http://www.bioinformatics.babraham.ac.uk/projects/trim_galore/) to enhance the chance of mapping. The resulting trimmed reads were then mapped again with bowtie. Any remaining unmapped and trimmed reads were processed with FLASH (*Magoč and Salzberg, 2011*) (version 1.2.8, RRID: SCR_005531, settings -m 9 -x 0.125) which combines overlapping read pairs and reconstructs read pairs without overlap. These are then realigned a third time using bowtie. PCR duplicates are then removed from the mapped bam files using samtools rmdup function (*Li et al., 2009*). Additionally, all reads overlapping any of the 'unmappable' UCSC Duke blacklisted hg19 regions (*ENCODE Project Consortium, 2012*) are also removed from the final bam file.

Open chromatin peaks were called through the aforementioned in-house pipeline by applying sample-specific read depth and width parameters, which were chosen based on the signal to noise ratio of a given sample.

## ChIP-seq data and identification of chromatin states
### Processing of available ChIP-seq data
Human islet ChIP-seq histone mark and TFBS data were obtained from various sources: H3K4me1, *CTCF* and H3K27ac (*Pasquali et al., 2014*), H3K36me3 and H3K4me3 (*Morán et al., 2012*) and H3K27me3 (*Kundaje et al., 2015*). Available raw fastq files were aligned to hg19 using bowtie1 (version 1.1.1) with modified default settings (-m 1 which removes reads with more than one valid alignment and -n 1 which defines the maximum number of mismatches in seed) and PCR duplicates were removed from the aligned bam files with Picard tools (RRID:SCR_006525, v1.119, http://broadinstitute.github.io/picard/). The resulting reads were converted into bed format using the bedtools bamToBed function (*Quinlan, 2014*) (RRID:SCR_006646, version v2.21.0) and extended by 200 bp towards the 3' end to account for fragment size.

### Identification of chromatin states using chromHMM
Binarised 200 bp density maps from the bed files of the 6 ChIP-seq marks were created using a Poisson distribution implemented in the BinaryBed function of the ChromHMM software as described in (*Ernst and Kellis, 2012*; *Ernst et al., 2011*). From these epigenomic density maps, 11 ChIP-only chromatin states were derived using a multivariate Hidden Markov Model implemented in the Learnmodel function (standard settings, h19 genome) of the software ChromHMM (*Ernst and Kellis, 2012*).

To generate additional sets of chromatin states based on ChIP-seq, ATAC-seq and DNA methylation data, ATAC-seq open chromatin and DNA methylation status were binarised. Specifically, ATAC-seq peaks (presence/absence) and whole-genome CpG methylation status (hypermethylation/ hypomethylation based on a threshold of 60% methylation) were binarised across 200 bp windows of the genome.

These binarised 200 bp ChIP-seq, ATAC-seq and DNA methylation maps were combined and used to generate 3 sets of chromatin states derived from ChIP and DNA methylation data (ChIP +Meth), ChIP and ATAC-seq data (ChIP +ATAC) or ChIP, ATAC-seq and DNA methylation data (ChIP +ATAC + Meth) using the Learnmodel ChromHMM function (*Figure 3A* and *Figure 3—figure supplement 1A–B*). As suggested by (*Ernst et al., 2011*), after evaluating models with up to 20 chromatin states, a 15 state model was chosen based on the resolution provided by identified states

### ADCY5 capture C analysis and ATAC-seq in EndoC-ßH1

Next-generation Capture-C was performed in order to map physical chromatin interactions with the *ADCY5* promoter in EndoC-ßH1 (RRID:CVCL_L909) cell lines (n = 3) (see protocol in Materials and methods in [*Davies et al., 2016*]).

In brief, chromatin conformation capture (3C) libraries were generated by formaldehyde fixation prior to DpnII restriction enzyme digestion and subsequent DNA ligation. Following cross-link reversal, DNA extraction and sonication, sequencing adapters were added to sonicated fragments (~200 bp). Library fragments were subjected to a double capture through hybridisation with a biotinylated oligonucleotide probes encompassing the *ADCY5* promoter and enriched using streptavidin bead pulldown. PCR amplified fragments were then sequenced through paired-end sequencing (Illumina Next-Seq). An in silico restriction enzyme digestion was performed on the set of reconstructed fragments (from paired-end sequenced reads) using the DpnII2E.pl script (*Davies, 2015*)(https://github.com/Hughes-Genome-Group/captureC). Uncaptured reads and PCR duplicates were removed prior to mapping to the human genome (hg19) with Bowtie (*Langmead et al., 2009*)(v 1.1.0). Removal of PCR duplicates and classification of fragments as 'capture' (i.e. including the *ADCY5* promoter) or 'reporter' (outside the capture fragment on exclusion region) was performed with the CCanalyser2. pl wrapper (*Davies, 2015*)(https://github.com/Hughes-Genome-Group/captureC). Unique mapped interactions were normalized to the total number of *cis* interactions (i.e. same chromosome) per 100,000 interactions. Significant chromatin interactions were determined from a rank-sum test implemented in the program peakC (*de Wit and Geeven, 2017*)(https://github.com/deWitLab/peakC). Specifically, we evaluated interactions involving all SNPs in high LD (r2 >0.8) with the lead rs11708067. The lead variant (rs11708067) was in high LD with 15 SNPs (mapping to 12 DpnII fragments) that spanned a region of 47 kb. We applied the Benjamini-Hochberg correction to control the false discovery rate for the set of p-values corresponding to each restriction fragment within the 47 kb region at the ADCY5 locus.

In addition, ATAC-seq was performed in 50,000 cells of EndoC-ßH1 cell lines (n = 3) and the data was analysed in the same way as described above for human islet samples.

Endo-βH1 cells were obtained from Endocells and have been previously authenticated (*Ravassard et al., 2011*). In addition, the cell line was tested and found negative for mycoplasma contamination.

### Overlaying generated epigenomic datasets generated here with other genomic regulatory regions

CpG sites and/or hypomethylated regulatory regions identified from the WGBS and/or 450 k array data were overlapped with existing islet chromatin state maps (*Parker et al., 2013*), islet transcription factor binding sites (*FOXA2, MAFB, NKX2.2, NKX6.1, PDX1*), T2D-associated islet dDMRs (*Dayeh et al., 2014*) and eQTLs (*van de Bunt et al., 2015*). Similarly, ATAC-seq open chromatin peaks generated here were overlapped with publicly available ATAC-seq peaks (*Varshney et al., 2017*).

In addition, we also obtained the 850 k array manifest file to determine overlap of 850 k array CpG sites with GWAS credible set regions (https://support.illumina.com/downloads/infinium-methylationepic-v1-0-product-files.html).

### Genetic datasets used in this study

Credible sets from the DIAGRAM (*Scott et al., 2017*)(involving 26.7 k cases and 132.5 k controls of predominantly European origin, imputed to the 1000G March 2012 reference panel) and ENGAGE (*Horikoshi et al., 2015*)(including 46.7 k individuals, imputed to the 1000G March 2012 reference

panel) consortium were used to compare the ability of the 450 k, 850 k and WGBS methylation array to interrogate T2D and FG GWAS regions.

The DIAGRAM and ENGAGE GWAS SNP summary level data was used for the FGWAS analysis to determine enrichment of regulatory annotations in T2D and FG GWAS signal.

Furthermore, data from (*Wood et al., 2017*) and (*Dimas et al., 2014*) were used to categories T2D GWAS loci into physiological groups of insulin secretion, insulin resistance or unclassified loci.

## Statistical and computational analysis

## Enrichment analysis of identified regulatory annotations in other genomic annotations

Enrichment of hypomethylated regulatory regions (LMRs and UMRs, result section 2.2.) and ATAC-seq open chromatin peaks (result section 2.3) in the aforementioned genomic annotations (method section 4.6) was determined through 100,000 random permutations. P-values and fold enrichment was determined by comparing the true overlap results to the permuted overlap results. The resulting P-values were multiple testing corrected using Bonferroni correction (an adjusted p-value<0.05 was considered significant).

## FGWAS enrichment analysis

FGWAS (*Pickrell, 2014*) (version 0.3.6) applied a hierarchical model that determined shared properties of loci affecting a trait. The FGWAS model used SNP-based GWAS summary level data and divided the genome into windows (setting 'k'=5000 which represents the number of SNPs per window), which are larger than the expected LD patterns in the population. The model assumed that each window either contained a single SNP that affected the trait or that there was no SNP in the window that influenced the trait. The model estimated the prior probability of a window to contain an association and the conditional prior probability that a SNP within the window was the causal variant. These prior probabilities were variable, dependent on regional annotations and estimated based on enrichment patterns of annotations across the genome using a Bayes approach.

### FGWAS single state analysis

FGWAS was used with standard settings to determine enrichment of individual islet chromatin states, LMRs, UMRs, PMDS and ATAC-seq open chromatin peaks, CDS and CONS sequence in DIA-GRAM (setting 'cc' was applied for use with T2D-case-control GWAS data) and ENGAGE GWAS SNP summary level data.

For each individual annotation, the model provided maximum likelihood enrichment parameters and annotations were considered as significantly enriched if the parameter estimate and 95% CI was above zero.

### FGWAS joint model analysis

To determine the maximum likelihood model the following approach suggested by (*Pickrell, 2014*) was used for each set of chromatin states (ChIP-only, ChIP +ATAC, ChIP +Meth and ChIP +ATAC + Meth), separately. In addition, CDS and CONS sequenced were used as well for each set of chromatin states in the joint analysis. Firstly, a model was fitted for each annotation individually to identify all annotations that were significantly enriched with the trait. Secondly, the annotation with the highest increase (and enrichment) in the maximum log-likelihood was added to the model and the analysis was repeated with the additional annotation. Thirdly, annotations were added as long as they increase the maximum log-likelihood of the newly tested model. Fourthly, a 10-fold cross-validation approach was used after determining a penalty parameter based on the maximum likelihood of a penalised log-likelihood function to avoid overfitting. Fifthly, each annotation was dropped from the model and the maximum cross-validation likelihood was evaluated. If a reduced model has a higher cross-validation maximum likelihood, additional annotations are dropped until the model cannot be further improved. This model was described as the best fitted model and used for the remaining analysis. The maximum likelihood enrichment parameters and 95% CI for each annotation of the best model were reported (independent of significance).

## Comparing FGWAS enrichment parameter across chromatin states

Initially, similar enhancer chromatin states derived from the four different ChromHMM analyses (ChIP-only, ChIP + ATAC, ChIP + Meth, ChIP + ATAC + Meth) were compared. Similarity was determined based on shared histone chromatin marks according to the chromHMM emission parameters. Further comparisons between the ChIP-only and ChIP + ATAC + Meth model were performed based on the reweighted FGWAS maximum variant PPA and the number of reweighted 99% credible set variants per T2D locus (for details regarding FGWAS PPA see next section).

However, considering that the chromatin states were derived from distinct sets of annotations across different analyses of ChromHMM, a direct comparison was not fully possible. Hence, a nested model approach was used to further dissect the contribution of open chromatin and DNA methylation to the enrichment. Specifically, an FGWAS analysis was performed that combined the ChIP-only chromHMM states with raw LMRs (representing DNA methylation) and ATAC-seq peaks (representing open chromatin). After determining the best maximum-likelihood cross-validation model (combining ChIP-only, ATAC-seq and LMR states) a nested model and log-likelihood ratio test were used to determine the contribution of each annotation to the model (*Figure 3—figure supplement 1D*).

## Reweighting of variant PPA and testing of allelic imbalance

The enrichment priors derived from the FGWAS maximum likelihood model were used as a basis for evaluating both the significance and functional impact of associated variants in GWAS regions; allowing variants that map into annotations that show global enrichment to be afforded extra weight.

Specifically, variants at significant GWAS regions with a high FGWAS PPA (PPA >= 10%) and overlapping open enhancer states were prioritised for further follow-up. Genome-wide significance of loci was determined based on P-values ($p<5\times10^{-8}$) or a regional FGWAS PPA >= 90% (representing the sum of the PPAs of all SNPs in a given region). The latter threshold is based on a recommendation from (*Pickrell, 2014*) who observed that a regional PPA of 90% or above can be used to identify sub-threshold GWAS loci.

Of the prioritised variants, only variants with at least two heterozygous samples and ATAC-seq read depth of at least nine reads (minimum five reads for each allele) were tested for allelic imbalance.

To avoid read-mapping and reference allele bias the software WASP (*van de Geijn et al., 2015*) (Version 0.2261) was used to remove reads associated with mapping bias. In short, reads of the unfiltered bam file that overlapped the variant of interest were identified. For each read overlapping an SNP, the genotype of that SNP was changed to the alternative allele and the read was remapped using bwa (*Li and Durbin, 2009*) (version 0.5.8 c). Any read that failed to realign in the same position in the genome was discarded. Ultimately, PCR duplicates were filtered using the WASP 'rmdup_pe.py' script, which removed duplicated reads randomly (independent of the mapping score) to avoid any bias.

Allelic imbalance was determined using a binomial test as implemented in R.

## Identification of TFBS at SNPs that display allelic imbalance

The tool 'Fimo'(*Grant et al., 2011*) implemented in the 'meme' software package (RRID:SCR_001783) was applied to identify TF motifs that significantly (FDR < 0.05) matched the sequence overlapping a SNP variant showing allelic imbalance (20 bp up and downstream).

## Overlap of regulatory regions

Overlap between genomic regulatory regions was performed using bedtools intersectBed function (*Quinlan, 2014*) (version 2.21.0). Summary statistics across 200 bp windows were determined using bedtools mapBed function. Random permutations of regulatory regions were performed by applying the bedtools shuffleBed function.

## Statistical analysis

All statistical analysis (unless otherwise stated) was performed using R (version 3.0.2) including Spearman's correlation analysis to compare the 450 k and WGBS array, the KS-test to compare 450 k and WGBS DNA methylation distributions, the binomial test to evaluate allelic imbalance and

principal component analysis to identify batch effects in the 450 k data. Significance is defined as p<0.05 unless otherwise stated.

## Visualisation and figure generation

All figures unless otherwise stated were generated using R (version 3.0.2) and/or ggplot2(*Wickham, 2009*). *Figure 1E* was generated using locuszoom (*Pruim et al., 2010*). Chromatin state CHiP-seq enrichment maps (*Figure 3A*, *Figure 3—figure supplement 1A–B*) were generated using chromHMM (*Ernst and Kellis, 2012*). The genome-browser views (*Figure 5*) were generated using the UCSC genome browser tool (*Kent et al., 2002*).

## Sequencing data

ATAC-seq and WGBS sequencing data has been deposited at the EBI hosted European Genome-phenome Archive (EGA, https://ega-archive.org/) and is accessible via the EGA accession numbers: EGAS00001002592, EGAD00001003946 and EGAD00001003947.

## Acknowledgements

We thank the High-Throughput Genomics Group at the Wellcome Centre for Human Genetics (funded by Wellcome grant reference 090532) for the generation of the Sequencing data. MT was supported by a Wellcome Doctoral Studentship. MvdB was supported by a Novo Nordisk postdoctoral fellowship run in partnership with the University of Oxford. JMT is supported by Wellcome Strategic Award. VN is funded by the European Union Horizon 2020 Programme (T2DSYSTEMS). SB was supported by a Royal Society Wolfson Research Merit Award (WM100023). ALG is a Wellcome Senior Fellow in Basic Biomedical Science (095101/Z/10/Z and 200837/Z/16/Z) and MIM is a Wellcome Senior Investigator. The research was supported by the National Institute for Health Research (NIHR) Oxford Biomedical Research Centre (BRC). This work was also supported by EU (HEALTH-F4-2007-201413), Wellcome (090367, 090532, 106130, 098381 and 099673/Z/12/Z) and NIH (U01-DK105535, U01-DK085545, R01-DK098032 and R01-MH090941) grants. The views expressed are those of the author(s) and not necessarily those of the NHS, the NIHR or the Department of Health.

## Additional information

### Competing interests

Mark I McCarthy: Senior editor, *eLife*. The other authors declare that no competing interests exist.

### Funding

| Funder | Grant reference number | Author |
| --- | --- | --- |
| Novo Nordisk | | Martijn van de Bunt |
| Horizon 2020 Framework Programme | HEALTH-F4-2007-201413 | Vibe Nylander<br>Anna L Gloyn |
| Royal Society | | Stephan Beck |
| National Institute for Health Research | | Anna L Gloyn<br>Mark I McCarthy |
| National Institutes of Health | U01-DK105535 | Anna L Gloyn<br>Mark I McCarthy |
| Wellcome | 099673/Z/12/Z | Matthias Thurner |
| National Institutes of Health | R01-DK098032 | Anna L Gloyn<br>Mark I McCarthy |
| Wellcome | 106130 | Jason M Torres<br>Anna L Gloyn<br>Mark I McCarthy |
| National Institutes of Health | R01-MH090941 | Anna L Gloyn<br>Mark I McCarthy |

| Wellcome | 095101/Z/10/Z | Anna L Gloyn |
| Wellcome | 200837/Z/16/Z | Anna L Gloyn |
| Wellcome Trust | 090532 | Mark I McCarthy<br>Anna L Gloyn |
| Wellcome | 098381 | Mark I McCarthy |
| National Institutes of Health | U01-DK085545 | Mark I McCarthy |
| Wellcome | 090367 | Mark I McCarthy |

The funders had no role in study design, data collection and interpretation, or the decision to submit the work for publication.

## Author contributions

Matthias Thurner, Conceptualization, Data curation, Software, Formal analysis, Validation, Investigation, Visualization, Methodology, Writing—original draft; Martijn van de Bunt, Kyle J Gaulton, Conceptualization, Methodology, Writing—review and editing; Jason M Torres, Formal analysis, Validation, Methodology, Writing—review and editing; Anubha Mahajan, Resources, Formal analysis, Supervision, Methodology, Writing—review and editing; Vibe Nylander, Data curation, Formal analysis, Validation, Investigation, Methodology, Writing—review and editing; Amanda J Bennett, Resources, Methodology, Writing—review and editing; Amy Barrett, Carla Burrows, Resources, Methodology; Christopher G Bell, Data curation, Writing—review and editing; Robert Lowe, Data curation, Methodology, Writing—review and editing; Stephan Beck, Conceptualization, Resources, Data curation, Writing—review and editing; Vardhman K Rakyan, Conceptualization, Resources, Data curation, Methodology, Writing—review and editing; Anna L Gloyn, Mark I McCarthy, Conceptualization, Resources, Supervision, Funding acquisition, Project administration, Writing—review and editing

## Author ORCIDs

Matthias Thurner (iD) http://orcid.org/0000-0001-7329-9769
Martijn van de Bunt (iD) http://orcid.org/0000-0002-6744-6125
Anna L Gloyn (iD) http://orcid.org/0000-0003-1205-1844
Mark I McCarthy (iD) http://orcid.org/0000-0002-4393-0510

## Ethics

Human subjects: The Human Research Ethics Board at the University of Alberta (Pro00001754), the University of Oxford's Oxford Tropical Research Ethics Committee (OxTREC Reference: 2-15), or the Oxfordshire Regional Ethics Committee B (REC reference: 09/H0605/2) approved the studies. All organ donors provided informed consent for use of pancreatic tissue in research.

## Decision letter and Author response

Decision letter https://doi.org/10.7554/eLife.31977.046
Author response https://doi.org/10.7554/eLife.31977.047

# Additional files

## Supplementary files

• Transparent reporting form
DOI: https://doi.org/10.7554/eLife.31977.024

## Major datasets

The following datasets were generated:

| Author(s) | Year | Dataset title | Dataset URL | Database, license, and accessibility information |
|---|---|---|---|---|
| Matthias Thurner, Anna L Gloyn, Mark I McCarthy | 2017 | Islet open chromatin data | https://ega-archive.org/datasets/EGAD00001003947 | Available through controlled access at the EGA website (study accession: EGAS00001002592 and dataset accession: EGAD 00001003947) |
| Matthias Thurner, Anna L Gloyn, Mark I McCarthy | 2017 | Islet DNA methylation data | https://ega-archive.org/datasets/EGAD00001003946 | Available through controlled access at the EGA website (study accession: EGAS00001002592 and dataset accession: EGAD00001003946) |

The following previously published datasets were used:

| Author(s) | Year | Dataset title | Dataset URL | Database, license, and accessibility information |
|---|---|---|---|---|
| Morán I, Akerman I, van de Bunt M, Xie R, Benazra M, Nammo T, Arnes L, Nakić N, García-Hurtado J, Rodríguez-Seguí S, Pasquali L, Sauty-Colace C, Beucher A, Scharfmann R, van Arensbergen J, Johnson PR, Berry A, Lee C, Harkins T, Gmyr V, Pattou F, Kerr-Conte J, Piemonti L, Berney T, Hanley N, Gloyn AL, Sussel L, Langman L, Brayman KL, Sander M, McCarthy MI, Ravassard P, Ferrer J | 2012 | Transcription profiling by high throughput sequencing of human and mouse pancreatic islet-cells | https://www.ebi.ac.uk/arrayexpress/experiments/E-MTAB-1294/ | Available from Array Express (accession no: E-MTAB-1294) |
| Parker SC, Stitzel ML, Taylor DL, Orozco JM, Erdos MR, Akiyama JA, van Bueren KL, Chines PS, Narisu N, NISC Comparative Sequencing Program, Black BL, Visel A, Pennacchio LA, Collins FS | 2013 | Chromatin stretch enhancer states drive cell-specific gene regulation and harbor human disease risk variants | https://www.ncbi.nlm.nih.gov/geo/query/acc.cgi?acc=GSE51312 | Available from NCBI GEO (accession no: GSE51312) |

| | | | | |
|---|---|---|---|---|
| Pasquali L, Gaulton KJ, Rodríguez-Seguí SA, Mularoni L, Miguel-Escalada I, Akerman I, Tena JJ, Morán I, Gómez-Marín C, van de Bunt M, Ponsa-Cobas J, Castro N, Nammo T, Cebola I, García-Hurtado J, Maestro MA, Pattou F, Piemonti L, Berney T, Gloyn AL, Ravassard P, Gómez-Skarmeta JL, Müller F, McCarthy MI, Ferrer J | 2014 | Pancreatic islet enhancer clusters enriched in type 2 diabetes risk-associated variants | https://www.ebi.ac.uk/arrayexpress/experiments/E-MTAB-1919/ | Available from Array Express (accession no: E-MTAB-1919) |
| Roadmap Epigenomics Consortium, Kundaje A, Meuleman W, Ernst J, Bilenky M, Yen A, Heravi-Moussavi A, Kheradpour P, Zhang Z, Wang J, Ziller MJ, Amin V, Whitaker JW, Schultz MD, Ward LD | 2015 | Integrative analysis of 111 reference human epigenomes | http://www.roadmapepigenomics.org/ | Available from Epigenome Roadmap website |

| | | | | |
|---|---|---|---|---|
| Horikoshi M, Magi R, van de Bunt M, Surakka I, Sarin AP, Mahajan A, Marullo L, Thorleifsson G, Hagg S, Hottenga JJ, Ladenvall C, Ried JS, Winkler TW, Willems SM, Pervjakova N, Esko T, Beekman M, Nelson CP, Willenborg C, Wiltshire S, Ferreira T, Fernandez J, Gaulton KJ, Steinthorsdottir V, Hamsten A, Magnusson PK, Willemsen G, Milaneschi Y, Robertson NR, Groves CJ, Bennett AJ, Lehtimaki T, Viikari JS, Rung J, Lyssenko V, Perola M, Heid IM, Herder C, Grallert H, Muller-Nurasyid M, Roden M, Hypponen E, Isaacs A, van Leeuwen EM, Karssen LC, Mihailov E, Houwing-Duistermaat JJ, de Craen AJ, Deelen J, Havulinna AS, Blades M, Hengstenberg C, Erdmann J, Schunkert H, Kaprio J, Tobin MD, Samani NJ, Lind L, Salomaa V, Lindgren CM, Slagboom PE, Metspalu A, van Duijn CM, Eriksson JG, Peters A, Gieger C, Jula A, Groop L, Raitakari OT, Power C, Penninx BW, de Geus E, Smit JH, Boomsma DI, Pedersen NL, Ingelsson E, Thorsteinsdottir U, Stefansson K, Ripatti S, Prokopenko I, McCarthy MI, Morris AP, ENGAGE Consortium | 2015 | Discovery and Fine-Mapping of Glycaemic and Obesity-Related Trait Loci Using High-Density Imputation | http://diagram-consortium.org/2015_ENGAGE_1KG/ | Available from the ENGAGE Consortium website |
| van de Bunt M, Manning Fox JE, Dai X, Barrett A, Grey C, Li L, Bennett AJ, Johnson PR, Rajotte RV, Gaulton KJ, Dermitzakis ET, MacDonald PE, McCarthy MI, Gloyn AL | 2015 | Transcript Expression Data from Human Islets Links Regulatory Signals from Genome-Wide Association Studies for Type 2 Diabetes and Glycemic Traits to Their Downstream Effectors | https://www.ebi.ac.uk/ega/studies/EGAS00001001265 | Available from the EGA website (study accession no: EGAS00001002592) |

 

| Scott RA, Scott LJ, Mägi R, Marullo L, Gaulton KJ, Kaakinen M, Pervjakova N, Pers TH, Johnson AD, Eicher JD, Jackson AU, Ferreira T, Lee Y, et al | 2017 | An Expanded Genome-Wide Association Study of Type 2 Diabetes in Europeans | http://diagram-consortium.org/ | Available from the Diagram Consortium website |
|---|---|---|---|---|
| Varshney A, Scott LJ, Welch RP, Erdos MR, Chines PS, Narisu N, Albanus RD, Orchard P, Wolford BN, Kursawe R, Vadlamudi S, Cannon ME, Didion JP, Hensley J, Kirilusha A; NISC Comparative Sequencing Program, Bonnycastle LL, Taylor DL, Watanabe R, Mohlke KL, Boehnke M, Collins FS, Parker SC, Stitzel ML | 2017 | Genetic regulatory signatures underlying islet gene expression and type 2 diabetes | https://www.ncbi.nlm.nih.gov/projects/gap/cgi-bin/study.cgi?study_id=phs001188.v1.p1 | Available through the dbGaP website (study accession no: phs001188.v1.p1) |

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
