## [Decision Letter]

Thank you for submitting your article "Integration of human pancreatic islet genomic data refines regulatory mechanisms at Type 2 Diabetes susceptibility loci" for consideration by *eLife*. Your article has been reviewed by two peer reviewers, and the evaluation has been overseen by a Reviewing Editor and Aviv Regev as the Senior Editor. The reviewers have opted to remain anonymous.

The reviewers have discussed the reviews with one another and the Reviewing Editor has drafted this decision to help you prepare a revised submission.

The reviewers of your manuscript found that the publication of a comprehensive epigenomic map of human islets is of broad interest and a timely resource for the community. The study has significant strengths, including the important observation of the far superior performance of WGBS compared to even the more recent, expanded 800K methylation array in comprehensively annotating methylation sites across the genome. Another significant strength highlighted by all reviewers is the integration of this dataset with other functional annotations and genetic fine mapping for T2D-associted loci, readily identifying a subset of variants that are putatively causal of the associations, and laying out a general analytical strategy that can be used by others, beyond those with immediate interest in T2D genetics. Overall, the reviewers found that this study significantly contributes to the effort of generating comprehensive functional annotations of a tissue of great medical and biological importance and thus far not properly characterized yet.

Below is a summary of essential points to be addressed:

1) The data presented starting in Figure 2 and shown throughout the manuscript suggest that virtually all enrichments are significant using FDR (masked by reporting only "FDR < 0.05" in main text). A more appropriate representation would either be quantitative (size proportional to log10p) or using a more stringent (Bonferroni) correction. The enrichments shown could be depicted in a more quantitative manner.

2) Causal GWAS variants can be enriched in regions with other functional annotations, such as coding regions or regions with strong evolutionary conservation. Such regions have not been considered here. The authors chose to reweight their PPAs solely using chromatin data, but this would down-weight causal variants with other functional annotations in the same region.

3) How much have the segmentations from the HMM improved with the additional data generated (without using GWAS enrichment as the benchmark)? It appears as though FGWAS enrichments for the actual annotations themselves (shown in Figure 3—figure supplement 1 and Figure 3—figure supplement 2) are just as good, if not better, than the HMM analysis (could also look at simple non-HMM two-annotation enrichments). Another way to look at this in the authors' framework would be to compare the median # of credible set SNPs (and top PPA) using only these annotations for FGWAS.

4) Both the Bayes Factor and FGWAS approaches rely on the assumption that at most one SNP in each region is causal, which may not always be true. This limitation should be discussed.

5) The addition of the Capture-C data in Figure 5 does not appear to help resolve rs11708067 as the likely causal variant as a several LD variants are similarly (by eye) overlapping fragments of high Capture-C density. Could this be quantified?

6) Given the known advantages of Stratified LD Score Regression (Finucane et al., 2015) over FGWAS to uncover true enrichments, have the authors considered using this approach?

7) What proportion of the narrow sense heritability is captured by the PPA variants?

8) The nomenclature used to define individual loci emerging from GWAS has followed the historical approach of using the closest gene to the associated SNPs. It has recently been recognized that this is often an imprecise approximation, as pointed out by reviewer 1. While there is no expectation that the authors would come up with a solution to this nomenclature conundrum, it was felt that the authors might address this issue on the paper, highlighting that the gene names associated with each locus may not accurately reflect the gene target of the association.

9) It was also noted that, despite the depth and complexity of the analysis carried out in this work, key data are missing which would greatly enhance the value of this report. An example is provided by Figure 4: why not present a table giving the identified variants at each of the loci (or at least the top half?) with their causal probability? Presumably TCF7L2 refers to rs7903146? This should be clarified (as for the other loci included). It would also be nice to provide eQTL data for this and a few other chosen loci.

10) Finally, we would suggest that the authors consider bringing up the possibility that the association signal of several of the GWAS loci analyzed, including the ones highlighted in this work (see comment 5, above), might result not from the phenotypic impact of a single causal variant, but rather by the combined effects of multiple variants within the associated haplotype. This is certainly a topic that will become more obvious as we perform more detailed and high throughput computational and experimental analysis of GWAS loci, and it would be important for this work to come out already pondering about this possibility.

---

## [Author Response]

Below is a summary of essential points to be addressed:1) The data presented starting in Figure 2 and shown throughout the manuscript suggest that virtually all enrichments are significant using FDR (masked by reporting only "FDR < 0.05" in main text). A more appropriate representation would either be quantitative (size proportional to log10p) or using a more stringent (Bonferroni) correction. The enrichments shown could be depicted in a more quantitative manner.

Since, in the original analysis, only 1000 permutations had been performed, the lowest empirical P-value (0.001) did not lend itself to Bonferroni correction for the number of tests performed. Given the reviewers’ comment, we have increased the number of permutations to 100k to be able to estimate a more accurate empirical P-value and then applied a Bonferroni correction. This had no relevant impact on the findings since the empirical P-values calculated for most annotations were still less than 1/100,000 (equating to a minimum Bonferroni corrected P-value of 0.00032). All annotations that were significant using an FDR <0.05 remained significant using the Bonferroni correction (adjusted P<0.05) and the newly calculated enrichment estimates were in the same range as the original ones.

As requested we also represented the P-values more appropriately by providing the P-values in Figure 2 and the associated figure legend. In addition, we added Figure 2—figure supplement 1 that has been changed to show both the enrichment (x-axis) and associated –log10 Bonferroni adjusted P-value (y-axis) for a more quantitative presentation of the enrichment.

We updated the results in section 2.2 and modified the Materials and methods section 4.9.1 as well as Figure 2 and Figure 2—figure supplement 1.

2) Causal GWAS variants can be enriched in regions with other functional annotations, such as coding regions or regions with strong evolutionary conservation. Such regions have not been considered here. The authors chose to reweight their PPAs solely using chromatin data, but this would down-weight causal variants with other functional annotations in the same region.

We agree with the reviewers’ comments that it is important to include a wide range of functional annotations. In fact, and contrary to the reviewer comments, we had already included coding regions in the original FGWAS analysis.

In response to the reviewers’ suggestion, we have now repeated the analyses adding in conserved region annotation. We used conserved sequence annotations from Lindblad-Toh et al., 2011 and updated the results of the paper accordingly.

Consistent with previous observations (e.g. Finucane et al., 2015) we found that conserved sequence was significantly enriched for T2D association in the single-state analysis. However, in the joint model, conserved sequence annotation was *not* retained presumably because the signal arising from conserved sequence was adequately captured by the chromatin state and coding annotations. Therefore, the addition of conserved sequence annotation does not change the major findings and conclusions.

To include the updated analysis results the manuscript has been revised as follows: We included the updated results in Figure 3, Figure 4, Figure 3—figure supplement 1, Figure 3—figure supplement2 as well as Table 1–Table 2 and Figure 3—source data 1–Figure 3—source data 3. In addition, we updated the results described in the main text (in particular, section 2.3 Refining islet enhancer function using methylation and open chromatin data).

3) How much have the segmentations from the HMM improved with the additional data generated (without using GWAS enrichment as the benchmark)? It appears as though FGWAS enrichments for the actual annotations themselves (shown in Figure 3—figure supplement 1 and Figure 3—figure supplement 2) are just as good, if not better, than the HMM analysis (could also look at simple non-HMM two-annotation enrichments). Another way to look at this in the authors' framework would be to compare the median # of credible set SNPs (and top PPA) using only these annotations for FGWAS.

While high levels of enrichment do not necessarily reflect high levels of sensitivity across a large number of GWAS loci, we agree that this is an interesting question. To compare the effect of different annotations to the baseline model, we have generated the requested metrics (median number and distribution of 99% credible set SNPs and median top variant PPA and top variant PPA distributions) for the following datasets as suggested:

- ChIP-only

- ChIP+Meth

- ChIP+ATAC

- ChIP+ATAC+Meth

- ATAC-only

- LMR-only

As shown in the new Figure 4—source data 1 and Figure 4—figure supplement 1 (see below for additional information), we found that ATAC-seq on its own (ATAC-only) or in combination with ChIP-seq data (ChIP+ATAC), leads to a notable reduction in 99% credible set size and an increase in top variant PPA compared to the baseline ChIP-only model.

In contrast, DNA methylation did not improve fine-mapping resolution: DNA methylation on its own (LMR-only) or in combination with ChIP-seq data, actually led to a *reduction* in top variant PPA compared to the baseline ChIP-only model. Credible set size *increased* when only DNA methylation (LMR-only) was included in the FGWAS fine-mapping analysis.

Despite the opposing effects of ATAC-seq and DNA methylation, the combination of ChIP-seq, ATAC-seq and DNA methylation (ChIP+ATAC+Meth) also performed well. The ChIP+ATAC+Meth model showed the best results in terms of median credible set size, identified a higher number of significant segments/loci and was almost identical in performance to the ChIP+ATAC dataset in terms of median maximum variant PPA.

Overall these results indicate that the chromatin accessibility data provided by ATAC-Seq (either in its own or in combination with islet ChIP-seq annotations) has the most value for refining fine-mapping and homing in on likely causal variants at T2D GWAS loci.

These results are highly consistent with our original analysis described in section 2.3, which showed that the effect of open chromatin on genetic T2D risk predominates over the effects of DNA methylation. The additional results are shown in the new and updated Figure 4—figure supplement 1, Supplementary Figure 4—source data 1 and referred to in the main text:

“We also expanded the FGWAS PPA analysis to investigate open chromatin and DNA methylation effects on fine-mapping and found that the reduction in 99% credible set size and increase in maximum variant PPA was driven predominantly by open chromatin (Figure 4—figure supplement 1). This demonstrates that the inclusion of open chromatin maps helps to improve prioritisation of causal variants at many T2D GWAS loci.”

4) Both the Bayes Factor and FGWAS approaches rely on the assumption that at most one SNP in each region is causal, which may not always be true. This limitation should be discussed.

We agree with the reviewer. We have been working to implement additional analyses steps to circumvent this limitation. Specifically, we have been testing an adaptation to FGWAS that allows us to analyse regions with multiple signals, after conditional decomposition. This is something we hope to deploy in future analyses, but it is not yet ready for inclusion in this paper. Meantime, and in line with reviewers’ comment 10, we have included the following paragraph in the Discussion highlighting this limitation of FGWAS and how future studies could address this:

“Although we provide highly detailed functional fine-mapping of T2D genetic variants to uncover causal variants, the FGWAS approach applied in this study is limited in its ability to determine the effect of multiple variants at individual loci. […] Analysis methods that combine functional fine-mapping with conditional analysis and consider LD and haplotype patterns are likely to provide a more complete overview of the causal interactions at T2D GWAS loci.”

5) The addition of the Capture-C data in Figure 5 does not appear to help resolve rs11708067 as the likely causal variant as a several LD variants are similarly (by eye) overlapping fragments of high Capture-C density. Could this be quantified?

This is correct. In our experience using Capture-C, it is typical that promoter “baits” capture multiple (enhancer) sequences in the adjacent region. This is not unexpected given the strong correlation of functional activity within adjacent sequence (as reflected in TAD structure for example). We agree, therefore, that it is generally not possible to use capture-C in isolation to definitively resolve the mechanistic interactions at a GWAS signal. At the *ADCY5* locus, the capture-C data we present demonstrate physical approximation between rs11708067 and the *ADCY5* promoter in an appropriate cell type: this provides additional evidence (to be considered in tandem with other complementary data) to support their relationship. At *ADCY5*, we are able to push the interpretation a little further in terms of resolving the causal variant because rs11708067 is the sole variant in the enhancer region that shows this physical approximation to the promoter.

To support more rigorous assessments of Capture-C data, we have been implementing a method for quantitative detection of Capture-C peaks which takes into account the background distribution of interaction “noise”. We have quantified the evidence for interaction for a total of 12 fragments across the *ADCY5* region which contain rs11708067 or other close T2D-associated proxies. In this analysis, only two fragments (one of which contains rs11708067) are called as “peaks” on the basis of significant read number over background.

These updated results are shown in the newly added Figure 5—figure supplement 1 and in the main text as shown below:

“To resolve the significance of the interaction between the restriction fragment encompassing rs11708067 and the ADCY5 promoter, we used the programme peakC (de Wit and Geeven et al., 2017) (https://github.com/deWitLab/peakC) to evaluate the interactions of 12 fragments covering the lead SNP rs11708067 and 15 SNPs in high LD (r2 > 0.8) across a region of 47kb. […] These SNPs fall into a region that did not show evidence of open chromatin.”

In addition, we also added the following paragraph to the Materials and methods section to include the additional analysis:

“Significant chromatin interactions were determined from a rank-sum test implemented in the program peakC (de Wit and Geeven et al., 2017)(https://github.com/deWitLab/peakC). […] We applied the Benjamini-Hochberg correction to control the false discovery rate for the set of p-values corresponding to each restriction fragment within the 47kb region at the ADCY5 locus.”

6) Given the known advantages of Stratified LD Score Regression (Finucane et al., 2015) over FGWAS to uncover true enrichments, have the authors considered using this approach?

We agree with the authors that Stratified LD Score Regression could be a valuable method to run on our dataset, in particular, since it can include multiple causal signals at a locus. However, after considering the use of multiple methods (Garfield, Stratified LD Score regression, FGWAS) we ultimately decided to use FGWAS for the following reasons:

- The FGWAS framework provides a stringent and straightforward way to determine a maximum likelihood model with multiple annotations using both forward and reverse mode selection.

- The framework provides a simple way of calculating updated posterior probabilities based on the enrichment priors of the best maximum likelihood annotation model. Importantly, the joint maximum likelihood model provides relative enrichment estimates for each category that allow us to up-weight the PPA of variants overlapping a given annotation in a proportional way dependent on the annotation.

- FGWAS is not dependent on reference datasets for LD-scores and can use all variants tested in the 1000G imputed DIAGRAM dataset including low frequency and rare variants.

- The observed enrichment patterns (highest enrichment for active enhancer states) are highly consistent with previous studies (e.g. Parker et al., 2013, Pasquali et al., 2014 and Gaulton et al., 2015).

We will continue to review these different approaches and will seek to implement other methods – such as LD score regression – in future work.

7) What proportion of the narrow sense heritability is captured by the PPA variants?

We have estimated the narrow sense heritability for T2D using UK Biobank data from ~350k individuals. To avoid inflation of the heritability estimates by variants in high LD, we restricted the analysis to the (single) top PPA variant for each of the 52 segments that were significantly associated with T2D genome-wide. The narrow sense heritability for T2D was estimated at 1.8%. To provide a reference point, we also estimated that all currently published T2D GWAS variants (Scott et al., 2017, Mahajan et al., 2014, Morris et al., 2012, Voight et al., 2010) explain 5.7% of variation in T2D risk. We have also used the top associated FG variants at 20 genome-wide significant segments to calculate narrow sense heritability estimates for FG, using Oxford BioBank data from 3.6k individuals. The narrow sense heritability for FG was estimated at 1.9%.

8) The nomenclature used to define individual loci emerging from GWAS has followed the historical approach of using the closest gene to the associated SNPs. It has recently been recognized that this is often an imprecise approximation, as pointed out by reviewer 1. While there is no expectation that the authors would come up with a solution to this nomenclature conundrum, it was felt that the authors might address this issue on the paper, highlighting that the gene names associated with each locus may not accurately reflect the gene target of the association.

We agree! We have added the following sentence:

“(Of note, in line with traditional GWAS nomenclature, locus names were defined based on proximity between the lead variant and the closest gene and does not, of itself, indicate any causal role for the gene in T2D susceptibility).”

9) It was also noted that, despite the depth and complexity of the analysis carried out in this work, key data are missing which would greatly enhance the value of this report. An example is provided by Figure 4: why not present a table giving the identified variants at each of the loci (or at least the top half?) with their causal probability? Presumably TCF7L2 refers to rs7903146? This should be clarified (as for the other loci included). It would also be nice to provide eQTL data for this and a few other chosen loci.

- As requested we have created an additional table (Figure 4—source data 2) that provides information for all 190 variants from Figure 4 which overlap one of the annotations included in the FGWAS joint-model.

- Per variant, the following T2D/FGWAS information is provided: rsID; FGWAS PPA; T2D GWAS P-value; FGWAS segment number; T2D locus name; tested for allelic imbalance (Yes/No).

- If available, the following eQTL information (from Varshney et al., 2017) is also provided:

eQTL allele1 (effector), eQTL allele 2, eQTL q-value, eQTL effect and eQTL gene

10) Finally, we would suggest that the authors consider bringing up the possibility that the association signal of several of the GWAS loci analyzed, including the ones highlighted in this work (see comment 5, above), might result not from the phenotypic impact of a single causal variant, but rather by the combined effects of multiple variants within the associated haplotype. This is certainly a topic that will become more obvious as we perform more detailed and high throughput computational and experimental analysis of GWAS loci, and it would be important for this work to come out already pondering about this possibility.

In line with the reviewers’ comment 4 about the limitation of FGWAS to assume only single causal variants, we have added a paragraph in the Discussion describing the potential impact of multiple signals per locus and interacting causal variants on the phenotype and how this could be addressed. The part of the paragraph that highlights these effects is shown below:

“Specifically, FGWAS relies on the assumption of a single causal variant within each region, which may not necessarily be true for all loci. […] Analysis methods that combine functional fine-mapping with conditional analysis and consider LD and haplotype patterns are likely to provide a more complete overview of the causal interactions at T2D GWAS loci.”